# The continued influence of AI-generated deepfake videos despite transparency warnings
Simon Clark ✉ & Stephan Lewandowsky

Advances in artificial intelligence (AI) have made it easier to create highly realistic deepfake videos, which can appear to show someone doing or saying something they did not do or say. Deepfakes may present a threat to individuals and society: for example, deepfakes can be used to influence elections by discrediting political opponents. Psychological research shows that people's ability to detect deepfake videos varies considerably, making us potentially vulnerable to the influence of a video we have failed to identify as fake. However, little is yet known about the potential impact of a deepfake video that has been explicitly identified and flagged as fake. Examining this issue is important because current legislative initiatives to regulate AI emphasize transparency. We report three preregistered experiments ($N$ = 175, 275, 223), in which participants were shown a deepfake video of someone appearing to confess committing a crime or a moral transgression, preceded in some conditions by a warning stating that the video was a deepfake. Participants were then asked questions about the person's guilt, to examine the influence of the video's content. We found that most participants relied on the content of a deepfake video, even when they had been explicitly warned beforehand that it was fake, although alternative explanations for the video's influence, related to task framing, cannot be ruled out. This result was observed even with participants who indicated that they believed the warning and knew the video to be fake. Our findings suggest that transparency is insufficient to entirely negate the influence of deepfake videos, which has implications for legislators, policymakers, and regulators of online content.

In March 2022, weeks after the Russian invasion of Ukraine, a video being shared online appeared to show Ukraine's president, Volodymyr Zelensky, telling his people to surrender[1]. The video was an entirely fabricated 'deepfake' – so-called because these fake videos are produced using deep learning artificial intelligence (AI). Advances in AI mean that it is becoming ever easier to produce highly realistic deepfake videos, which can appear to show someone doing or saying something they did not, in reality, do or say. The Zelensky incident constituted a 'near miss' – largely because the video was a poor-quality deepfake and unlikely to fool many people. President Zelensky himself described the attempt to discredit him as a "childish provocation"[1]. But what if the quality of the video had been such that it could not be readily dismissed as a fake?

Although it is possible to create deepfakes for less nefarious purposes such as education and wellbeing[2,3], much of the scholarly discussion about deepfakes has focused on the more sinister implications of the technology: deepfake videos could be used to discredit or intimidate people[4], influence political opinion and elections[5], and undermine our criminal justice system by creating doubt about the reliability of video evidence in general, even when it is not fake[6]. These threats to society can arise through two distinct routes: First, people may not be able to detect that a video is fake and hence may be influenced by it. Second, in light of a large body of research showing that even overtly false or irrelevant information can influence people, e.g.,[7–9] it is possible for deepfakes to exert an influence even if they have been specifically identified and flagged as fake.

To date, most research has focused on the first route. It has been found that people's ability to detect deepfake videos varies considerably, depending largely on a particular video's quality and context. For some videos, groups of people can outperform the latest computer models[10], but deepfake videos, which are more difficult to spot can lead to low detection accuracy[11,12], particularly on a small screen such as a mobile phone[13]. Detection accuracy has also been related to demographics[14,15] and cognitive reflection abilities[16,17]. People tend to be overconfident in their ability to discern real

School of Psychological Science, University of Bristol, Bristol, UK. ✉e-mail: simon.clark@bristol.ac.uk

and fake videos[17,18] and less likely to spot a deepfake video when they agree with its content[13]. Further, deepfake videos will continue to become increasingly hard to discern from real videos as the technology improves[19,20]. People are therefore potentially more and more vulnerable to the influence of a video they have failed to identify as fake.

Here, we examine the second route to influence by asking what effects a deepfake video may have after it has already been identified and flagged as fake. Normatively, people should be expected to discount the content of a video they know to be fake – at least when forming beliefs about factual matters – because it has no evidential connection to reality. However, extant psychological research into the effects of misinformation casts doubt on people's ability to discount a deepfake: first, people have been shown to be influenced by discredited information they know to be false[21,22]; second, warnings (e.g., fact-check tags) are only partially effective at reducing belief in false information[23,24], with explicit 'rated false' warnings more effective than general warnings[25].

The relatively few psychological studies that deal specifically with warnings about deepfake videos provide mixed results: Some studies found generic 'media literacy' interventions to be partially effective, by reducing sharing intention (at least for those with lower susceptibility to believing misleading claims)[26] or reducing the impact of misinformation contained in the video[27,28]. Other studies found educational warnings about deepfakes to be ineffective at reducing deception[11,29] but effective at creating distrust in all videos, even when they are in fact real[11] or distrust in online news more generally[29].

These existing psychological studies do not adequately answer the question of whether people continue to be influenced by a flagged deepfake video, due to limitations in their experimental designs. Some of these studies did not include an influence measure at all[11,26], or employed a self-reported measure of the video's "credibility", without directly addressing its content[27]. Most of these studies were based on a satirical deepfake video, which was already in the public domain[17,27,29] and hence participants may have already been aware of or seen the video. It is also unclear whether participants' interpretation of and response to a satirical deepfake video can be generalized to serious deepfake videos. Only one of these studies included a real video for comparison with the deepfake[11], and none of them employed a true control condition – instead comparing participants who were told the video they watched was a deepfake, with those who watched the same deepfake but without the warning. These studies, therefore, had no baseline with which to compare the influence of a flagged deepfake video.

Our experiments go beyond these precedents in several important ways: First, we produced two deepfake videos of our own, specifically for these experiments, to eliminate the risk of familiarity with existing deepfakes already in the public domain. Second, we also produced two unedited real videos, using the same script and background as the deepfake videos, for direct comparison. Third, we produced a control version of one of our real videos, adding background sounds to obscure the relevant content. Fourth, we included a direct measure of the extent to which participants' judgement was influenced by the content of the video, and also gave participants the opportunity to explain their reasoning, in their own words. Fifth, we used two different videos to cover both the political and non-political domains.

The question of whether transparency (e.g., a warning stating that content is AI-generated) is sufficient to negate the influence of known deepfake videos is timely because legislation currently being rolled out to address the risks of AI-generated content relies heavily on transparency as a potential solution. For example, the European Union's Artificial Intelligence Act requires that: "Deployers of an AI system that generates or manipulates image, audio or video content constituting a deep fake, shall disclose that the content has been artificially generated or manipulated"[30]. An (often tacit) assumption underlying the emphasis on transparency is that it enables people to dismiss or disregard information, should it be in their interest to do so.

Across three preregistered experiments (at osf.io/sjw9h), we tested whether participants continued to rely on the content of a deepfake video,

even when they were specifically told beforehand that it was fake. We produced two new deepfake videos for this project: the first (used in Experiments 1 and 2) was produced using face swap software and featured a fictional local government official admitting that he had taken a bribe; the second (used in Experiment 3) was produced using generative-AI video software and featured a fictional vegan social media influencer confessing that she had been seen eating meat. This scenario was chosen to provide contrast with the video used in the first two experiments (i.e., non-political vs. political context, younger female vs. older male subject, moral transgression vs. serious crime). In all experiments, depending on the condition, participants either did or did not receive a transparency warning stating that the video was a deepfake before watching the video. In Experiment 2, we also tested a more generic warning that alerted participants to the existence of deepfake videos, without referring specifically to the video they were about to watch.

Because these experiments involved fictional scenarios in which a fictional character appeared in a fictional deepfake video, we relied on participants' ability to make a judgement within the narrative frame provided. We note that this 'nested fiction' structure may have introduced ambiguity in how participants interpreted the video as evidence – an issue to which we return in the discussion.

Based on existing research, we preregistered hypotheses anticipating that a specific warning shown before watching a video, stating that the video had been identified and flagged as a deepfake, would: (1) be partially, but not entirely, effective at reducing participants' reliance on the content of the video; (2) be partially, but not entirely, effective at convincing participants that the video is fake (even if it is real); and (3) be more effective in both these areas than a generic warning which highlights the existence of deepfakes generally, without referring specifically to the video they were about to watch. A full list of preregistered hypotheses can be found in Table 1.

## Methods
All Experiments were approved by the Research Ethics Committee of the School of Psychological Science at the University of Bristol, UK, under approval code 10874.

### Experiments 1 & 2
Both experiments were conducted as per our preregistrations (at osf.io/sjw9h). Experiment 1 was preregistered on 14 October 2022, and Experiment 2 was preregistered on 6 April 2023. There were no deviations from the preregistered protocol.

Target sample sizes were determined a priori using G*Power (v3.1) software, as detailed in our preregistrations. Participants were recruited from the United States, via the CloudResearch MTurk Toolkit, which has been shown to provide high-quality participants[31]. Experiment 1 participants were paid $1.25 to complete an eight-minute questionnaire, and Experiment 2 participants were paid $1.00 to complete a six-minute questionnaire, both on the Qualtrics platform. After exclusions, Experiment 1 had 175 participants, aged between 20 and 76 years ($M = 41.68$, $SD = 13.20$, 88 male, 84 female, 2 non-binary, 1 declined), and Experiment 2 had 275 participants, aged between 19 and 70 years ($M = 36.66$, $SD = 11.53$, 108 male, 162 female, 4 non-binary, 1 declined). Demographic data, including self-reported gender, were collected for descriptive purposes only and not included in our analyses, because our research questions and hypotheses concerned general cognitive and social processes, and experiments were not designed or powered to test demographic differences. No data on race or ethnicity were collected.

Data were excluded for participants who failed to complete any of the survey items for the dependent variables, or failed the attention check, or completed the survey in less than one minute longer than it took to watch the video. In experimental conditions, the attention check required participants to correctly answer the question: *"Where did he say the money is hidden?"*, which was revealed in the closing seconds of the video. From 222 total responses in Experiment 1, 47 participants were excluded: 18 for failing to complete the survey, and 29 for failing the attention check. From 338 total

**Table 1 | Analysis of preregistered hypotheses**

| Hypothesis | | | | Group means | | | t | (df) | p | d | [95% CI] | |
|---|---|---|---|---|---|---|---|---|---|---|---|---|
| **Perception of guilt** | | | | | | | | | | | | |
| *Experiment 1* | | | | | | | | | | | | |
| H1 | fake specific | > | control | 0.87 | > | −0.40 | 6.16 | 134.17 | **<0.001***** | 0.96 | 0.64 | 1.28 |
| *Experiment 2* | | | | | | | | | | | | |
| H1 | fake specific | > | control | 0.62 | > | −0.29 | 2.60 | 55.41 | **0.012*** | 0.60 | 0.14 | 1.05 |
| H1a | fake generic | > | fake specific | 0.73 | > | 0.62 | 0.26 | 70.98 | 0.799 | 0.06 | −0.40 | 0.52 |
| H1b | fake none | > | fake generic | 2.14 | > | 0.73 | 4.22 | 61.30 | **<0.001***** | 0.95 | 0.47 | 1.43 |
| H3a | real none | > | real generic | 2.18 | > | 1.49 | 2.62 | 62.55 | **0.011*** | 0.60 | 0.14 | 1.05 |
| H3b | real generic | > | real specific | 1.49 | > | 1.29 | 0.58 | 73.86 | 0.562 | 0.13 | −0.32 | 0.58 |
| *Experiment 3* | | | | | | | | | | | | |
| H1 | real none | > | fake specific | 2.00 | > | 1.04 | 2.74 | 88.65 | **0.007**** | 0.53 | 0.15 | 0.91 |
| H3 | real none | > | real specific | 2.00 | > | 1.56 | 1.57 | 114.82 | 0.119 | 0.29 | −0.07 | 0.65 |
| **Perception of fakeness** | | | | | | | | | | | | |
| *Experiment 1* | | | | | | | | | | | | |
| H2 | fake specific | > | control | 1.27 | > | −0.07 | 6.07 | 156.16 | **<0.001***** | 0.93 | 0.61 | 1.25 |
| *Experiment 2* | | | | | | | | | | | | |
| H2 | fake specific | > | control | 1.88 | > | 0.00 | 5.44 | 76.48 | **<0.001***** | 1.21 | 0.72 | 1.68 |
| H2a | fake specific | > | fake generic | 1.88 | > | 0.18 | 4.81 | 71.67 | **<0.001***** | 1.12 | 0.62 | 1.61 |
| H2b | fake generic | > | fake none | 0.18 | < | 0.27 | −0.25 | 73.64 | 0.801 | −0.06 | −0.51 | 0.39 |
| H4a | real generic | > | real none | −0.18 | < | 0.33 | −1.65 | 75.18 | 0.102 | −0.38 | −0.82 | 0.07 |
| H4b | real specific | > | real generic | 0.97 | > | −0.18 | 3.64 | 73.17 | **<0.001***** | 0.83 | 0.36 | 1.29 |
| *Experiment 3* | | | | | | | | | | | | |
| H2 | fake specific | > | real none | 0.76 | > | −0.60 | 4.13 | 97.90 | **<0.001***** | 0.80 | 0.40 | 1.19 |
| H4 | real specific | > | real none | −0.39 | > | −0.60 | 0.65 | 107.95 | 0.518 | 0.12 | −0.24 | 0.48 |

Significant *p*-values are shown in **bold** text, *p* <0.05*, *p* <0.01**, *p* <0.001***.

responses in Experiment 2, 63 participants were excluded: 14 for failing to complete the survey, and 49 for failing the attention check.

Both experiments employed a between-subjects design, with participants randomly allocated to one of the following conditions: Experiment 1 was an initial, simplified design with just two conditions: *control* (*n* = 100) and *fake specific* (*n* = 75). Experiment 2 employed a 2 (*real, fake* video) × 3 (*none, generic, specific* warning) factorial design, plus a *control* condition, making seven conditions in total: *control* (*n* = 48), *real none* (*n* = 39), *real generic* (*n* = 39), *real specific* (*n* = 38), *fake none* (*n* = 37), *fake generic* (*n* = 40), and *fake specific* (*n* = 34).

The experimental design and procedure are shown in Fig. 1. In both experiments, after providing informed consent and demographic information, participants were briefly introduced to a fictional local government official, John Carter, and then shown a 60-second video in which he appeared to have been secretly filmed in a bar, admitting to a crime (taking a bribe). Three versions of the video were used: the first was an unedited real version in which John Carter was played by an actor; the second was a deepfake version using the same script as the real video but with the actor's face swapped for that of another actor using DeepFaceLab software; the third was a version of the real video in which background sounds (from a slot machine in the bar) were used to obscure incriminating verbal content, which served as a control. Screenshots of *real*, *fake*, and *control* videos are shown in Fig. 2, and a transcript can be found in the Supplementary Information.

Before watching the video, participants were given either no warning at all, the following generic warning: *"WARNING: Please be aware of the existence of deepfake videos"*, or the following specific warning: *"WARNING: You should be aware that this video has been identified and flagged as a deepfake video"*. Both warnings were followed by a brief explanation for the

benefit of participants unfamiliar with the term 'deepfake', as follows: *"Deepfakes are doctored videos which appear to show someone doing or saying something they did not, in reality, do or say. Advances in machine learning mean it is becoming ever easier to produce highly-realistic deepfake videos."*

After watching the video, participants were asked *"Do you think John Carter is guilty of a crime?"* to assess perception of guilt, and *"Do you think the video is a deepfake?"* to assess perception of fakeness, both measured using a 7-point Likert scale (−3 = *definitely not*, +3 = *definitely yes*), as shown in Fig. 1. The midpoint was labelled *impossible to say* to most accurately reflect the normative response to a lack of evidence.

Participants were also given the opportunity to provide qualitative responses for perception of guilt, *"Please explain briefly why you answered [e.g., probably yes] to whether you think John Carter is guilty of a crime"*, and perception of fakeness, *"Please explain briefly why you answered [e.g., probably not] to whether you think the video is a deepfake"*, in their own words.

Finally, participants were thoroughly debriefed about the purpose of the experiment. Participants in the *real specific* condition, who had been told that a real video was a deepfake, were shown a different debrief that revealed this misdirection.

## Experiment 3

Experiment 3 was conducted as per our preregistration (at osf.io/sjw9h) of 23 September 2024. There were no deviations from the preregistered protocol.

Target sample sizes were determined a priori using G*Power (v3.1) software, as detailed in our preregistration. Participants were recruited from the United Kingdom via Prolific and paid £1.00 to complete a six-minute

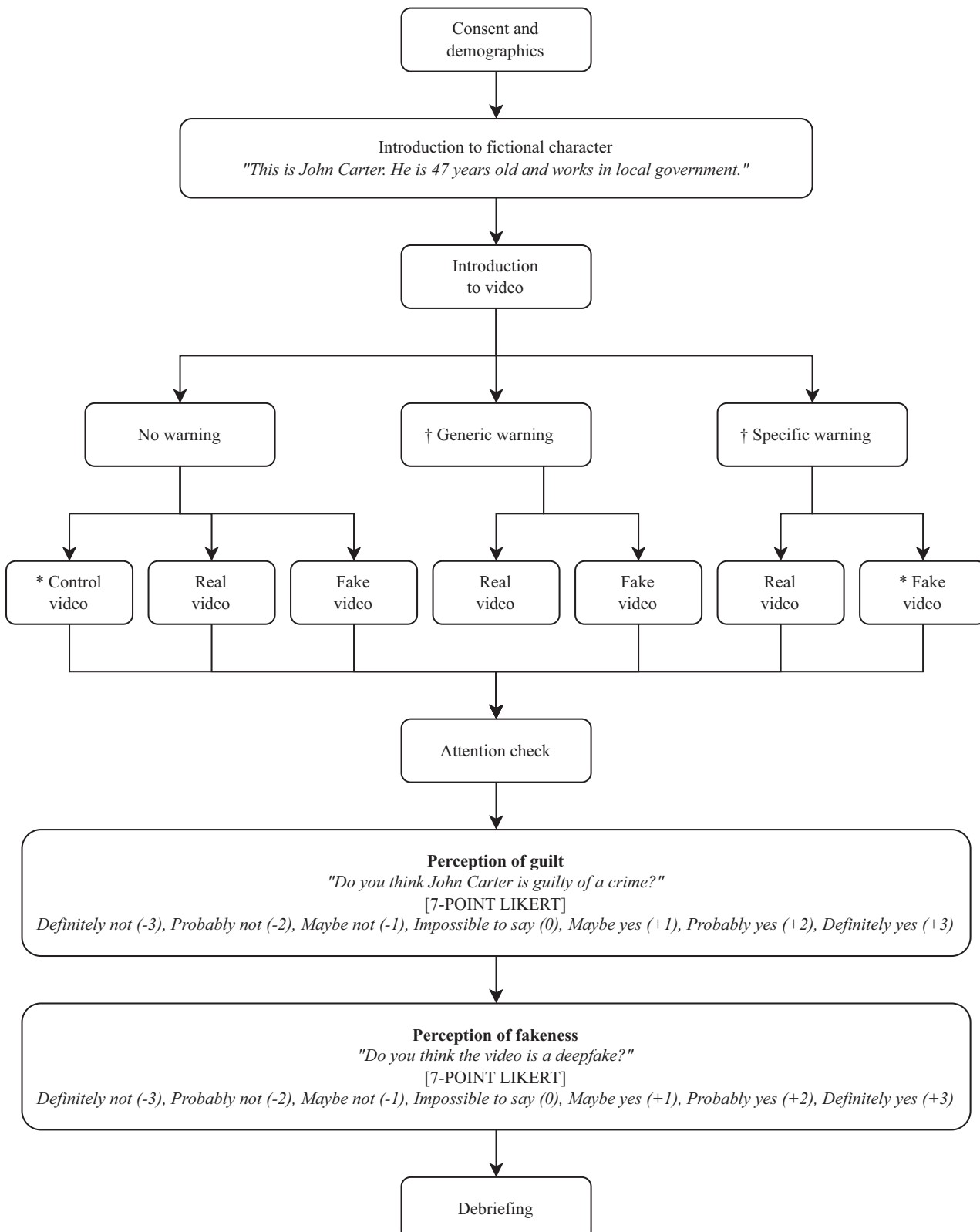

**Fig. 1 | Design & Procedure (Experiments 1 and 2).** * Experiment 1 used only those conditions marked with an asterisk and included two additional questions (see Method). † Both generic and specific warnings were followed by a brief explanation for the benefit of participants unfamiliar with the term 'deepfake', as follows:

*"Deepfakes are doctored videos which appear to show someone doing or saying something they did not, in reality, do or say. Advances in machine learning mean it is becoming ever easier to produce highly-realistic deepfake videos."*

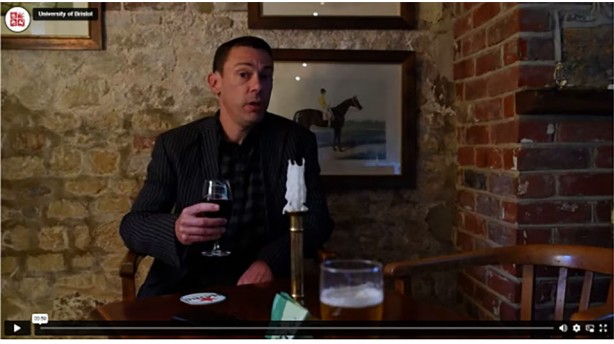

*Real* and *Control* videos (Experiments 1 & 2)

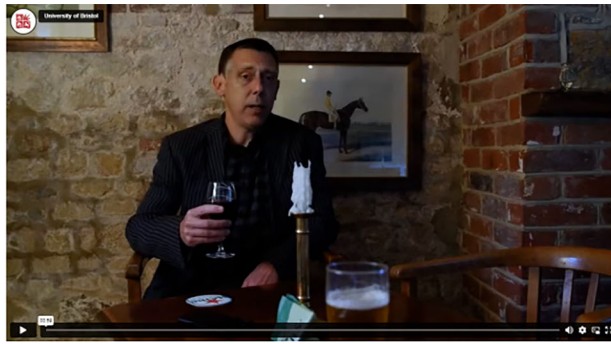

*Fake* video (Experiments 1 & 2)

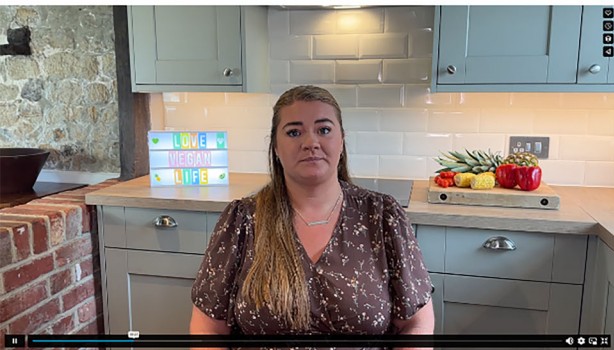

*Real* video (Experiment 3)

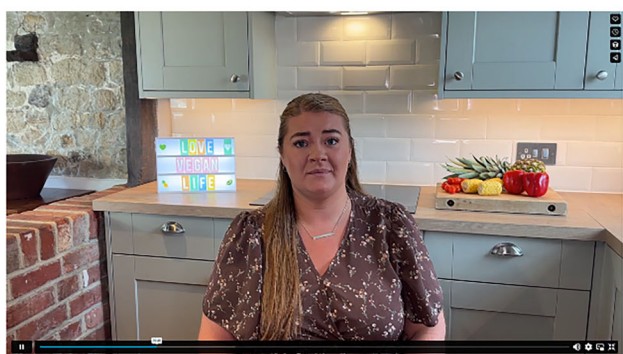

*Fake* video (Experiment 3)

**Fig. 2 | Screenshots of real (left) and deepfake (right) videos.** The control video used in Experiments 1 and 2 was based on the real video, but with added background sounds to obscure incriminating verbal content. Full transcripts of each video can be found in the Supplementary Information. The authors have obtained consent from the actors to publish these images.

questionnaire on the Qualtrics platform. After exclusions, Experiment 3 had 223 participants, aged between 19 and 58 years ($M = 30.54$, $SD = 8.71$, 99 male, 120 female, 3 non-binary, 1 declined). As in Experiments 1 and 2, demographic data were collected for descriptive purposes only and not included in our analyses.

As in Experiments 1 and 2, data were excluded for participants who failed to complete any of the survey items for the dependent variables, or failed the attention check, or completed the survey in less than one minute longer than it took to watch the video. The attention check required participants to correctly answer the question: *"Where did the incident described in the video take place?"* From the 238 total responses in Experiment 3, all participants completed the survey, but 15 participants were excluded for failing the attention check.

Experiment 3 employed a 2 (real, fake video) × 2 (none, specific warning) between-subjects factorial design, with participants randomly allocated to one of the conditions as follows: *real none* ($n = 60$), *real specific* ($n = 57$), *fake none* ($n = 56$), and *fake specific* ($n = 50$).

The experimental design and procedure were broadly the same as Experiments 1 and 2, as shown in Fig. 3. After providing informed consent and demographic information, participants were briefly introduced to a fictional social media influencer, Amelia Palmer, and were then shown a two-minute video in which she appeared to make a confession to her followers about a moral transgression (eating meat). Two versions of the video were used: the first was a real version in which Amelia was played by an actor; the second was a deepfake version created from scratch using generative-AI technology, trained using a single three-minute video of the same actor telling an unrelated story. Both versions used the same script, which can be found in the Supplementary Information. Screenshots of *real* and *fake* videos for Experiment 3 are also shown in Fig. 2.

Before watching the video, participants were given either no warning at all or the same specific warning used in Experiments 1 and 2, followed by an updated explanation for the benefit of participants unfamiliar with the term 'deepfake', as follows: *"Deepfakes are created using artificial intelligence (AI) and can appear to show someone doing or saying something they did not in reality do or say."*

After watching the video, participants were asked, *"Do you believe that vegan influencer Amelia Palmer was seen eating a burger?"* to assess perception of guilt, and *"Do you believe that the video is a deepfake?"* to assess perception of fakeness, both measured with the same 7-point Likert scale used in Experiments 1 and 2. Participants were also given the opportunity to briefly explain their reasons for each answer, in their own words, as in Experiments 1 and 2.

Finally, participants were thoroughly debriefed about the purpose of the experiment. Participants in the *real specific* condition, who had been told that a real video was a deepfake, were shown a different debrief, which revealed this misdirection.

**Data analysis**

Quantitative analysis was conducted as per our preregistrations (at osf.io/sjw9h), using JASP (v0.13). Two-sided Welch's *t*-tests were used throughout, because they are robust to violations of the homogeneity of variance assumption[32], which was breached in some conditions (as assessed using Levene's tests). Assumptions of normality were assessed using Shapiro-Wilk tests and were generally met; where minor deviations occurred, the tests were considered robust given the sample sizes employed. In Experiments 2 and 3, factorial effects were analysed using two-way between-subjects ANOVAs, followed by planned contrasts using Welch's *t*-tests. No corrections for multiple comparisons were applied, as each test addressed a

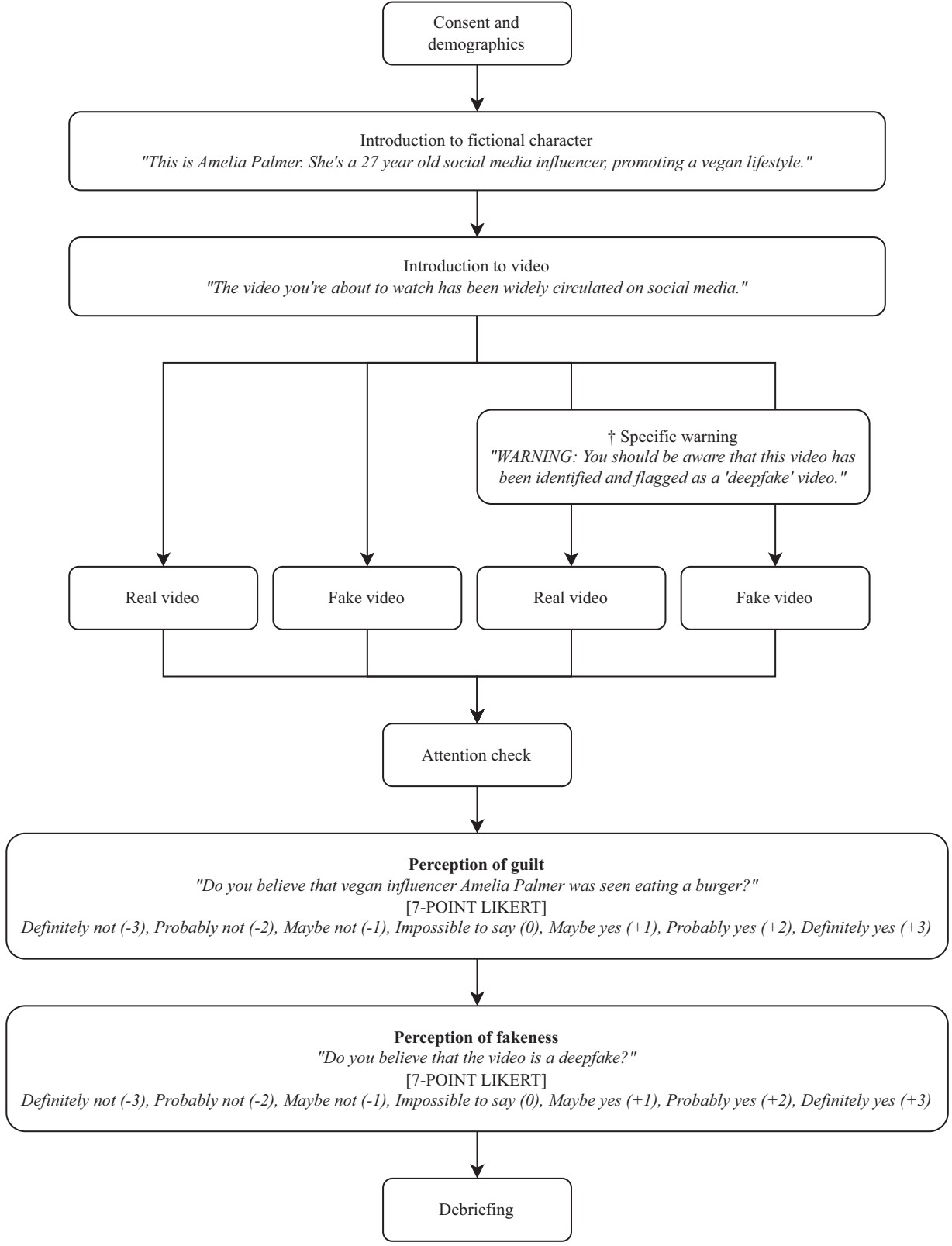

**Fig. 3 | Design & Procedure (Experiment 3).** † Specific warning was followed by an updated explanation for the benefit of participants unfamiliar with the term 'deepfake', as follows: *"Deepfakes are created using artificial intelligence (AI) and can appear to show someone doing or saying something they did not in reality do or say."*

distinct, preregistered hypothesis. Effect sizes are reported as Cohen's *d* for t-tests, calculated from the *t*-statistic and group sample sizes, and as partial eta squared ($\eta^2_p$) for ANOVAs, calculated from the sums of squares in the ANOVA output. Confidence intervals for effect sizes are reported where applicable.

Qualitative coding was conducted by one of the authors, plus a second independent rater. Participants were asked to provide explanations, in their own words, for their quantitative responses to the perception of guilt and perception of fakeness measures. These qualitative responses were coded based on whether the explanation

**Fig. 4 | Mean perception of guilt by condition.**
Error bars show 95% confidence intervals. Violin
plots and data points show the distribution of
responses. Experiment 1 was an initial, simplified
design with just two conditions: *control* ($n = 100$),
and *fake specific* ($n = 75$). Experiment 2 employed a 2
×3 factorial design, plus a control condition, thus:
*control* ($n = 48$), *real none* ($n = 39$), *fake none*
($n = 37$), *real generic* ($n = 39$), *fake generic* ($n = 40$),
*real specific* ($n = 38$), and *fake specific* ($n = 34$).
Experiment 3 employed a 2 ×2 between-subjects
factorial design, thus: *real none* ($n = 60$), *fake none*
($n = 56$), *real specific* ($n = 57$), and *fake spe-*
*cific* ($n = 50$).

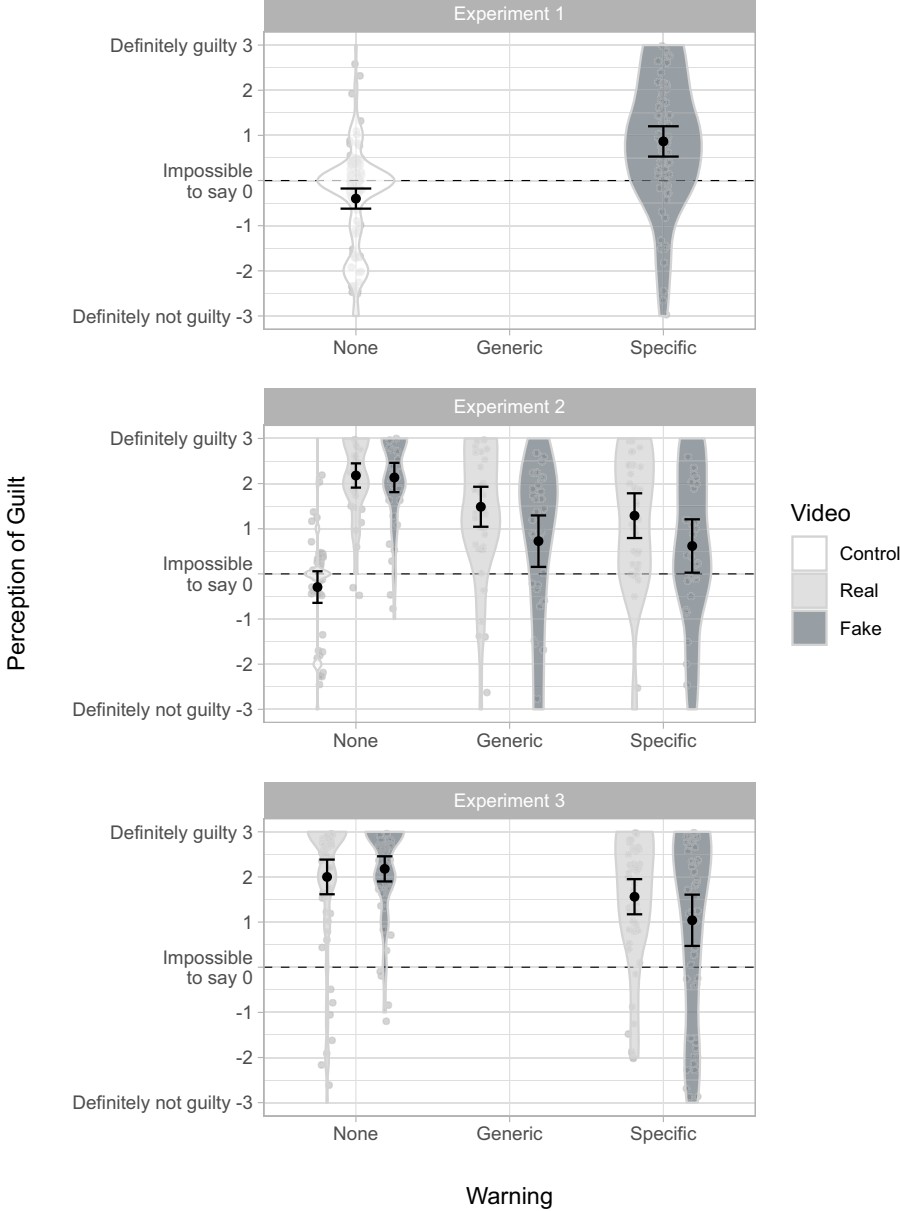

related to the warning shown before the video (coded W), the video
itself (coded V), both warning and video (coded WV), or neither
(coded NO). Initial coding achieved inter-rater reliability, measured
using Cohen's kappa, of K = 0.86, which is interpreted as almost
perfect agreement[33]. Disagreements were resolved for the final data
set by discussion between the two raters. Ratings were then combined
with a simplified version of the corresponding quantitative data
(*maybe, probably, definitely* aggregated) to give a single measure
describing both a participant's judgement and their reason for
making that judgement. For example, the response *definitely yes / "He
admitted that he took the money"* was rated *guilty, because of video*
(coded G/V) whereas the response *maybe not / "I can't say based on a
fake video"* was rated *not guilty, because of warning* (coded N/W).
Full coding data can be found at osf.io/sjw9h.

### Reporting summary

Further information on research design is available in the Nature Portfolio
Reporting Summary linked to this article.

## Results
### Experiment 1
Experiment 1 was an initial, simplified design with just two between-
subjects conditions: a *fake specific* condition, in which participants were
shown a deepfake video and specifically warned beforehand that it was fake,
and a *control* condition, in which participants were shown the control video,
as described above, with no warning beforehand. Participants in the *control*
condition were therefore provided with no specific information to assess
John Carter's guilt or innocence. Data for Experiment 1 were collected on
24–25 November 2022.

Perception of guilt ratings by condition are shown in Fig. 4, and ana-
lysis of associated hypotheses is shown in Table 1. Perception of guilt was
greater in the *fake specific* condition (0.87; close to *maybe yes*) than in the
*control* condition (−0.40; near *impossible to say*), $t(134.17) = 6.16$, $p < 0.001$,
$d = 0.96$, 95% CI [0.64, 1.28] (H1).

Qualitative responses for perception of guilt are summarized in
Table 2. This shows that 53.3% ($n = 40$, coded G/V or G/WV) of the 75
participants in the *fake specific* condition believed John Carter was guilty

**Table 2 | Summary of qualitative results, showing the percentage of participants in each condition who believed the person was guilty, based on the video**

| Video | Warning | | |
|---|---|---|---|
| | **None** | **Generic** | **Specific** |
| Experiment 1 | | | |
| Fake | | | 53.3% |
| | | | (n = 40 / 75) |
| Control | 12.0% | | |
| | (n = 12 / 100) | | |
| Experiment 2 | | | |
| Fake | 86.5% | 62.5% | 47.1% |
| | (n = 32 / 37) | (n = 25 / 40) | (n = 16 / 34) |
| Real | 89.7% | 82.1% | 65.8% |
| | (n = 35 / 39) | (n = 32 / 39) | (n = 25 / 38) |
| Control | 10.4% | | |
| | (n = 5 / 48) | | |
| Experiment 3 | | | |
| Fake | 87.5% | | 56.0% |
| | (n = 49 / 56) | | (n = 28 / 50) |
| Real | 83.3% | | 78.9% |
| | (n = 50 / 60) | | (n = 45 / 57) |

Participants are included if they gave a positive rating of guilt, and explained this rating with reference to the video's content (coded G/V or G/WV).

based (entirely or partially) on the content of the video, compared with 12.0% (n = 12) of the 100 participants in the *control* condition. Only 13.3% (n = 10, coded N/W or I/W) of participants in the *fake specific* condition cited the warning they had been shown beforehand, stating that the video was a deepfake, as their sole reason for answering *not guilty* or *impossible to say*.

Perception of fakeness ratings by condition are shown in Fig. 5, and analysis of associated hypotheses is shown in Table 1. Perception of fakeness was greater in the *fake specific* condition (1.27; *maybe yes*) than in the *control* condition (–0.07; *impossible to say*), $t(156.16) = 6.07, p < 0.001, d = 0.93$, 95% CI [0.61, 1.25] (H2).

Qualitative responses for perception of fakeness indicated that, of the 75 participants in the *fake specific* condition, 69.3% (n = 52) believed the video was a deepfake. Only 42.3% (n = 22) of these 52 participants mentioned the warning they were given before watching the video as a reason for believing it was a deepfake, whereas 61.5% (n = 32) referred to some aspect of the video itself (e.g., visual defects).

We also conducted a conditional analysis for the subset of 52 participants who were shown a *specific* warning followed by the deepfake video, and subsequently indicated that they believed the warning and therefore knew the video to be fake. Mean perception of guilt for this subset of participants who accepted the warning (0.87) was still greater than in the *control* condition (–0.40), $t(84.74) = 5.50, p < 0.001, d = 0.97$, 95% CI [0.61, 1.34]. Qualitative responses indicated that, despite believing the warning, 53.8% (n = 28) of these participants nevertheless relied on the content of the video to conclude that John Carter was guilty, compared with 12.0% (n = 12) of 100 participants in the *control* condition.

Two additional variables were measured in Experiment 1: Perception of suitability was measured with the question *"Do you think John Carter is a suitable person to hold a public position (e.g., planning department, tax office)?"*, using the same 7-point Likert scale, reverse-coded to show unsuitability. Mean perception of unsuitability was greater in the *fake specific* condition (1.45; *maybe yes*) than in the *control* condition (0.13; *impossible to say*), $t(151.57) = 6.61, p < 0.001, d = 1.02$, 95% CI [0.70, 1.34].

Perception of authenticity was measured with the question *"Do you think the video shows what actually happened?"*, using the same Likert scale, reverse-coded to show inauthenticity. There was no statistically significant difference between conditions for mean perception of inauthenticity, $t(155.54) = 0.68, p = .0496, d = 0.11$, 95% CI [–0.40, 0.20].

## Experiment 2

Experiment 2 employed a 2 (*real, fake* video) × 3 (*none, generic, specific* warning) between-subjects factorial design, plus a *control* condition as in Experiment 1, making seven conditions in total. Data for Experiment 2 were collected on 13–14 April 2023.

Perception of guilt ratings by condition are shown in Fig. 4, and analysis of associated hypotheses is shown in Table 1. There was a significant main effect for both the video variable, $F(1,221) = 6.55, p = 0.011, \eta^2_P = 0.029$, 95% CI [0.001, 0.085], and the warning variable, $F(2,221) = 15.41, p < 0.001, \eta^2_P = 0.122$, 95% CI [0.050, 0.203], with no significant interaction, $F(2,221) = 1.39, p = 0.252, \eta^2_P = 0.012$, 95% CI [0.000, 0.050]. As in Experiment 1, mean perception of guilt was greater in the *fake specific* condition (0.62; *maybe yes*) than in the *control* condition (–0.29; *impossible to say*), $t(55.41) = 2.60, p = 0.012, d = 0.60$, 95% CI [0.14, 1.05] (H1). We found no statistically significant difference between *fake generic* and *fake specific* conditions, $t(70.98) = 0.26, p = 0.799, d = 0.06$, 95% CI [–0.40, 0.52] (H1a), but mean perception of guilt was greater in the *fake none* condition (2.14; *probably yes*) than in the *fake generic* condition (0.73; *maybe yes*), $t(61.30) = 4.22, p < 0.001, d = 0.95$, 95% CI [0.47, 1.43] (H1b), and similarly, mean perception of guilt was greater in the *real none* condition (2.18; *probably yes*) than in the *real generic* condition (1.49, *maybe yes*), $t(62.55) = 2.62, p = 0.011, d = 0.60$, 95% CI [0.14, 1.05] (H3a). We found no statistically significant difference between *real generic* and *real specific* conditions, $t(73.86) = 0.58, p = 0.562, d = 0.13$, 95% CI [–0.32, 0.58] (H3b).

Qualitative responses for perception of guilt are summarized in Table 2. This shows that 89.7% (n = 35) of the 39 participants in the *real none* condition, who were shown a real video of John Carter admitting to committing a crime without any prior warning, believed him to be guilty based on the video, compared with 10.4% (n = 5) of 48 participants in the *control* condition. In the *fake generic* condition, 62.5% (n = 25) of 40 participants believed John Carter to be guilty based on the video, reducing further to 47.1% (n = 16) of 34 participants in the *fake specific* condition.

Perception of fakeness ratings by condition are shown in Fig. 5, and analysis of associated hypotheses is shown in Table 1. There was a significant main effect for both the video variable, $F(1,221) = 4.04, p = 0.046, \eta^2_P = .018$, 95% CI [0.000, 0.067] and the warning variable, $F(2,221) = 18.74, p < 0.001, \eta^2_P = 0.145$, 95% CI [0.067, 0.228], with no significant interaction, $F(2,221) = 1.95, p = 0.145, \eta^2_P = 0.017$, 95% CI [0.000, 0.060]. As in Experiment 1, mean perception of fakeness was greater in the *fake specific* condition (1.88; *probably yes*) than in the *control* condition (0.00; *impossible to say*), $t(76.48) = 5.44, p < 0.001, d = 1.21$, 95% CI [0.72, 1.68] (H2). Mean perception of fakeness was greater in the *fake specific* condition (1.88; *probably yes*) than in the *fake generic* condition (0.18; *impossible to say*), $t(71.67), = 4.81, p < 0.001, d = 1.12$, 95% CI [0.62, 1.61] (H2a), but we found no statistically significant difference between *fake generic* and *fake none* conditions, $t(73.64) = –0.25, p = 0.801, d = –0.06$, 95% CI [–0.51, 0.39] (H2b), nor between *real generic* and *real none* conditions, $t(75.18) = –1.65, p = 0.102, d = –0.38$, 95% CI [–0.82, 0.07] (H4a). Mean perception of fakeness was greater in the *real specific* condition (0.97; *maybe yes*) than in the *real generic* condition (–0.18, *impossible to say*), $t(73.17) = 3.64, p < 0.001, d = 0.83$, 95% CI [0.36, 1.29] (H4b).

Qualitative responses for perception of fakeness indicated that, whilst 82% (n = 28) of the 34 participants in the *fake specific* condition believed the video was a deepfake, only 43% (n = 12) of these 28 participants mentioned the warning they were given before watching the video, whereas 71% (n = 20) referred to some aspect of the video itself (e.g., visual defects).

We also conducted a conditional analysis for the subset of 47 participants who were shown a warning (either *generic* or *specific*), followed by the

**Fig. 5 | Mean perception of fakeness by condition.**
Error bars show 95% confidence intervals. Violin plots and data points show the distribution of responses. Experiment 1 was an initial, simplified design with just two conditions: *control* ($n = 100$), and *fake specific* ($n = 75$). Experiment 2 employed a 2 × 3 factorial design, plus a control condition, thus: *control* ($n = 48$), *real none* ($n = 39$), *fake none* ($n = 37$), *real generic* ($n = 39$), *fake generic* ($n = 40$), *real specific* ($n = 38$), and *fake specific* ($n = 34$). Experiment 3 employed a 2 × 2 between-subjects factorial design, thus: *real none* ($n = 60$), *fake none* ($n = 56$), *real specific* ($n = 57$), and *fake specific* ($n = 50$).

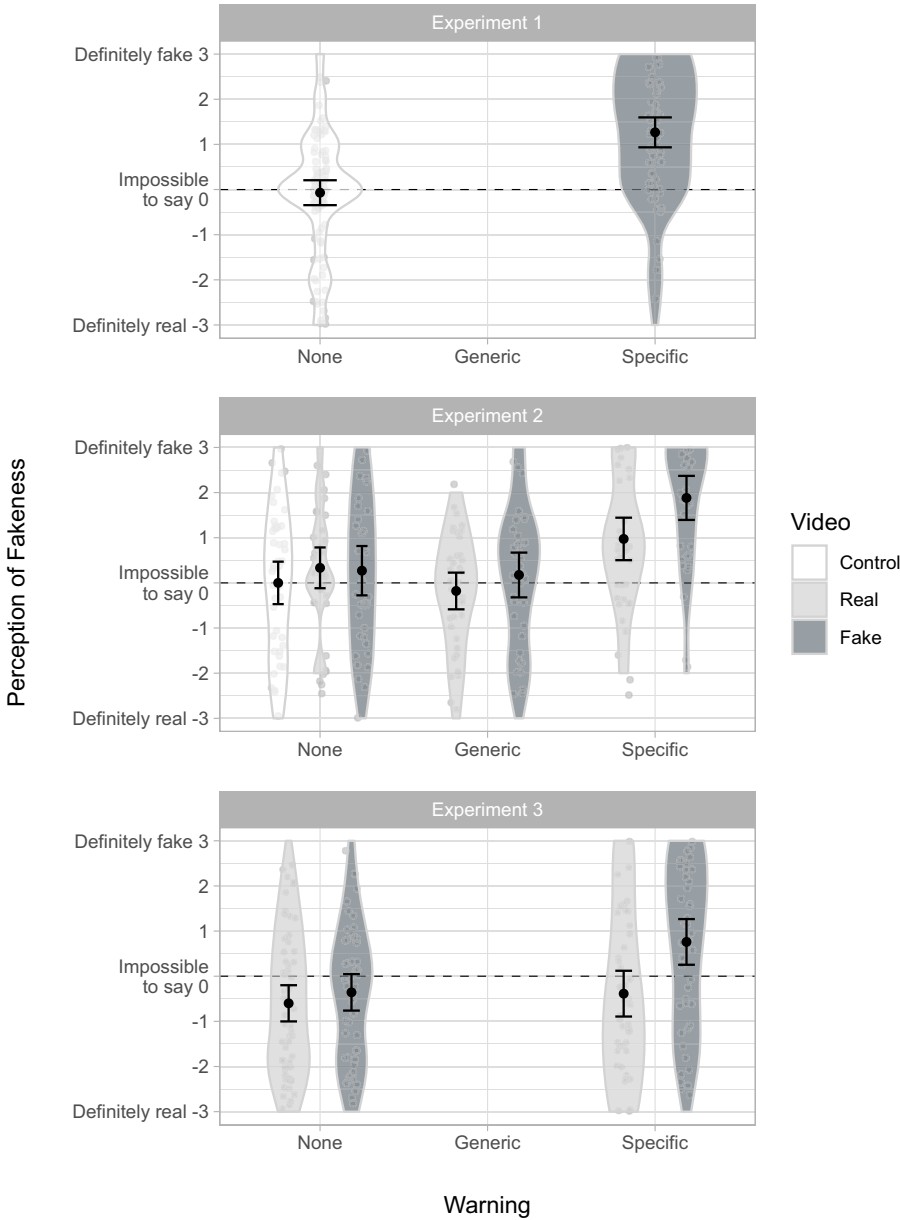

deepfake video, and subsequently indicated that they believed the warning and therefore knew the video to be fake. Mean perception of guilt for this subset of participants who accepted the warning (0.43) was still greater than in the *control* condition (−0.29), $t(77.52) = 2.13$, $p = 0.036$, $d = 0.44$, 95% CI [0.03, 0.85]. Qualitative responses indicated that, despite believing the warning, 44.7% ($n = 21$) of this subset of participants nevertheless relied on the content of the video to conclude that John Carter was guilty, compared with 10.4% ($n = 5$) of 48 participants in the *control* condition.

## Experiment 3

Experiment 3 employed a 2 (*real, fake* video) × 2 (*none, specific* warning) between-subjects factorial design, based on a new AI-generated deepfake video. Data for Experiment 3 were collected on 30 September 2024.

Perception of guilt ratings by condition are shown in Fig. 4, and analysis of associated hypotheses is shown in Table 1. There was a significant main effect for the warning variable, $F(1,219) = 14.26$, $p < 0.001$, $\eta^2_p = 0.061$, 95% CI [0.014, 0.132], but not the video variable, $F(1,219) = 0.68$, $p = 0.413$, $\eta^2_p = 0.003$, 95% CI [0.000, 0.034]. As in Experiments 1 and 2, mean perception of guilt was greater than zero in the *fake specific* condition (1.04; *maybe yes*), but lower than in the *real none* condition (2.00; *probably yes*), $t(88.65) = 2.74$, $p = .007$, $d = 0.53$, 95% CI [0.15, 0.91] (H1).

Qualitative responses for perception of guilt are summarized in Table 2. This shows that 83.3% ($n = 50$) of 60 participants in the *real none* condition, and 87.5% ($n = 49$) of 56 participants in the *fake none* condition, believed Amelia Palmer to be guilty based on the video, reducing to 56.0% ($n = 28$) of 50 participants in the *fake specific* condition.

Perception of fakeness ratings by condition are shown in Fig. 5, and analysis of associated hypotheses is shown in Table 1. There was a significant main effect for both the video variable, $F(1,219) = 8.94$, $p = 0.003$, $\eta^2_p = 0.039$, 95% CI [0.005, 0.101], and the warning variable, $F(1,219) = 8.21$, $p = 0.005$, $\eta^2_p = 0.036$, 95% CI [0.004, 0.097], with no significant interaction, $F(1,219) = 3.78$, $p = 0.053$, $\eta^2_p = 0.017$, 95% CI [0.000, 0.065]. As in Experiments 1 and 2, mean perception of fakeness was greater in the *fake specific* condition (0.76; *maybe yes*) than in the *real none* condition (−0.60; *maybe not*), $t(97.90) = 4.13$, $p < 0.001$, $d = 0.80$, 95% CI [0.40, 1.19] (H2).

Qualitative responses for perception of fakeness indicated that, whilst 60% ($n = 30$) of the 50 participants in the *fake specific* condition believed the video was a deepfake, only 20% ($n = 6$) of these 30 participants mentioned

the warning they were given before watching the video, whereas 80% ($n = 24$) referred to some aspect of the video itself (e.g., visual defects).

We also conducted a conditional analysis for the subset of 30 participants who were shown a *specific* warning followed by the deepfake video, and subsequently indicated that they believed the warning and therefore knew the video to be fake. Mean perception of guilt for this subset of participants who accepted the warning was still greater than zero (0.67), but lower than in the *real none* condition (2.00), $t(42.83) = 2.95$, $p = 0.005$, $d = 0.69$, 95% CI [0.23, 1.16]. Qualitative responses indicated that, despite believing the warning, 50.0% ($n = 15$) of these participants nevertheless relied on the content of the video to conclude that Amelia Palmer was guilty, compared with 83.3% ($n = 50$) of 60 participants in the *real none* condition.

## Discussion

Across three preregistered experiments, we found that: (a) Watching a deepfake video of someone appearing to make an admission of guilt influenced participants' beliefs about that person's guilt. This influence was observed, albeit with a smaller effect size, even for participants who were shown a specific warning beforehand stating that the video had been identified and flagged as a deepfake. Further, this influence was observed even among the subset of participants who explicitly stated that they believed the warning and knew the video was fake. (b) Participants who were shown the specific warning were (unsurprisingly) more likely to believe the video was a deepfake, whereas a generic warning – which alerted participants more generally to the existence of deepfakes – had no effect on participants' stated belief that the video they watched was a deepfake. (c) Despite having no influence on their belief that the video was a deepfake, the generic warning nevertheless reduced participants' perception of guilt.

These findings were consistent across three experiments and two very different videos. Figure 4 illustrates the striking similarity between the results of Experiments 2 and 3 despite substantial differences between the style and content of the videos used in each experiment (political vs. non-political, serious crime vs. trivial moral transgression, older male vs. younger female subject), and even the technology used to produce them (face swap vs. generative AI).

Our findings add to a growing body of research showing that transparency is not enough to entirely negate the influence of AI-generated content. For example, recent studies have found warnings had no significant effect on the persuasiveness of microtargeted political messages[34], or the extent to which participants followed moral advice generated by a large language model[35]. Relatedly, a recent study found that labels informing users that online content was AI-generated reduced belief in that content, but had little effect on users' stated likelihood of engaging with it[36], suggesting that such labels may not meaningfully alter behaviour. This was demonstrated by our conditional analyses, examining only those participants who believed our specific warning and therefore knew that the video they watched was a deepfake. Despite knowing the video was a deepfake, this group nevertheless continued to rely (entirely or partially) on the content of the video when making a judgement about the person's guilt. Transparency warnings can therefore be seen as a useful tool for reducing (to some extent) the influence of a known deepfake video, but transparency alone is by no means a complete solution to the various threats associated with deepfake videos.

Further, our findings suggest that a generic warning may alter people's interpretation of the content of a video, even when they remain unconvinced that the video is a deepfake. This is perhaps explained by what the authors of an earlier deepfake study termed "generalized uncertainty", finding that an educational warning about deepfake videos contributed to cynicism and distrust of social media news more generally[29]. Similarly, a thematic analysis of tweets about deepfakes related to the invasion of Ukraine found that such tweets indicated reduced trust in video content generally[37]. Moral philosophers have described awareness of deepfake videos as an epistemic threat to the credibility of video as evidence of reality[38,39], which presents an obvious risk to the use of video evidence in court[6,40]. To avoid contributing to a generalized cynicism about the authenticity of videos, we therefore suggest avoiding altogether the use of generic warnings about deepfake videos.

The specific warning used in Experiment 2 was partially effective at reducing perception of guilt, and also partially effective at convincing participants that the video they watched was a deepfake – even when it was, in fact, real. This finding supports existing literature across several disciplines arguing that malicious actors could take advantage of what has been termed the "liar's dividend" – i.e., claiming that an inconvenient video is a deepfake, when in fact it is real, e.g.[5,41]. Furthermore, incorrectly labelling real videos as fake could contribute to the generalized cynicism and lack of trust in videos discussed above. It is therefore important to be certain that a video is indeed a deepfake before labelling it as such.

Finally, our findings support the argument that some people employ a 'seeing is believing' heuristic to make a personal judgement about the authenticity of a video[16], even when they have been specifically warned that it is fake. Our qualitative responses show that a majority of participants who were shown a deepfake video, and were warned beforehand that it was fake, nevertheless commented on some aspect of the video itself (e.g., technical defects) to explain why they believed it to be fake, rather than simply stating that they had been told it was fake. This suggests a lack of trust in transparency warnings, or at the very least a lack of trust in our specific warning, which appeared to have been given by the social media platform. A future study might explore how the source of such a warning (e.g., content creator, social media platform, independent fact checker, police, government) might influence trust in its accuracy.

The perception of authenticity measure used in Experiment 1 was intended to test whether participants believed the video had been manipulated, but qualitative responses indicated that the question was widely misunderstood. Many participants referred to the sound effects which had been added to obscure incriminating content in the *control* condition (e.g., *definitely yes / "Most of it was bleeped out"*). Whilst such responses perhaps signalled media literacy, this was impossible to isolate using this measure. Some participants stated that a description of something happening is not the event itself (e.g., *probably yes / "It may have told what happened but not shown"*). Other participants answered in relation to the accuracy of what was said (e.g., *maybe not / "While he said he accepted money I am not clear exactly what actually happened"*). We did not consider the results of this variable to be meaningful, given the various ways in which the question was interpreted by participants.

## Limitations

We acknowledge a common limitation of psychological experiments, namely that our findings may have been influenced by demand characteristics, which occur when participants attempt to anticipate the purpose of an experiment and respond accordingly[42]. For example, a participant might choose to ignore the warning if they believed that the experimenters were trying to trick them into believing a real video was a deepfake. However, we do not believe that demand characteristics played a significant role in our findings, for two reasons: First, our qualitative data did not provide any hints that participants' reasoning was based on anticipating the purpose of the experiment – such post-experiment enquiry is the most obvious way to detect demand characteristics[42]. Second, we included a manipulation check by asking whether participants believed that the video they watched was a deepfake. This showed that even the subset of participants in the *fake specific* condition who believed the warning – and therefore knew that the video they were watching was a deepfake – were nevertheless influenced by the content of the video.

A second limitation common to experiments employing a fictional scenario is the possibility that participants may not respond as they would in a real-world situation[43]. It is widely understood, however, that people can make moral judgements about fictional characters in movies. For example, few viewers would conclude that a character is innocent of his on-screen crimes solely on the basis that the film is fictional. This has been termed a 'fictive pass', allowing participants to evaluate the actions of fictional characters much as they would in the real world[44]. Participants in the present study were likewise asked to assess guilt based on the content of the video they viewed, regardless of its fictional nature.

Our experiments involved the added complication of fictional scenarios in which a fictional character appeared in a fictional deepfake video, and we note that this 'nested fiction' structure may have introduced ambiguity in how participants interpreted the video as evidence. We therefore cannot rule out the possibility that our deepfake videos' influence may partly reflect inattention to the epistemic structure of the task, rather than pure susceptibility to non-probative information. Distinguishing between these two alternative interpretations is an important direction for future research, which should assess participants' meta-awareness of the fictional scenario, in addition to standard attention checks.

However, our qualitative data provided no indication that this limitation affected participants' reasoning in the present study. Participants' free-text explanations engaged directly with the in-scenario evidence and its credibility, and none suggested uncertainty about the nested fictional structure.

## Conclusions

Our findings have implications for legislators, policy makers, and regulators of social media platforms and online news. We have shown how the impact of a malicious deepfake video was not entirely negated by a warning beforehand stating that it was fake. Our perception of suitability measure illustrates the potential for real-world consequences: participants' beliefs about John Carter's suitability for public office were influenced by the deepfake video, despite the warning. This undermines regulators' current focus on transparency, which is seen as central to mitigating the risks of AI[30,45], despite there being little empirical evidence to support the effectiveness of AI transparency[35]. Specifically, our results indicate that it is insufficient to identify and flag deepfake videos that have been published online. Further measures, such as removing or prohibiting deepfake content, should therefore be considered.

We have thus shown that warnings will not adequately prevent the personal, political, and societal harms that could arise from the publication of a deepfake video in which a person's words or actions are misrepresented. Malicious actors could therefore use deepfake videos to discredit a business rival or political opponent, knowing that this strategy will be effective to some degree even if the video is immediately identified and flagged as fake. This represents a significant threat that demands the urgent attention of researchers and regulators alike.

## Data availability

Preregistrations and full reproduction data, for both quantitative and qualitative responses, are available at osf.io/sjw9h. This is an Open Access article distributed under the terms of the Creative Commons Attribution License (https://creativecommons.org/licenses/by/4.0/), which permits unrestricted reuse, distribution, and reproduction in any medium, provided the original work is properly cited.

## Code availability

No custom code was used; all analyses were conducted using standard functionality in JASP (v0.13).

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

## Acknowledgements

S.L. acknowledges financial support from the European Research Council (ERC Advanced Grant 101020961 PRODEMINFO), the Humboldt Foundation through a research award, the Volkswagen Foundation ('Reclaiming individual autonomy and democratic discourse online: How to rebalance human and algorithmic decision making' grant), and the European Commission (Horizon 2020 grant 101094752 SoMe4Dem and UKRI EU Horizon replacement funding grant number 10049415). The funders had no role in study design, data collection and analysis, decision to publish or preparation of the manuscript.

## Author contributions

S.C. and S.L. conceptualized and designed the research and collected the data. S.C. analysed the data and wrote the manuscript. S.L. edited and contributed to the manuscript.

## Competing interests

The authors declare no competing interests.
