## [Transparent Peer Review file · Communications Psychology]

The Continued Influence of AI-Generated Deepfake Videos Despite Transparency Warnings

Corresponding Author: Mr Simon Clark

Version 0:

Decision Letter:

Dear Mr Clark,

Thank you for your patience during the peer-review process. Your manuscript titled "Seeing is Believing: The Continued Influence of Known AI-Generated 'Deepfake' Videos" has now been seen by 3 reviewers, and I include their comments at the end of this message. They find your work of interest but raised some important points. We are interested in the possibility of publishing your study in Communications Psychology, but would like to consider your responses to these concerns and assess a revised manuscript before we make a final decision on publication.

We therefore invite you to revise and resubmit your manuscript, along with a point-by-point response to the reviewers. Please highlight all changes in the manuscript text file.

Editorially, we consider it crucial that evidence to support the data quality for online participants, such as attention checks or additional experimental evidence that the findings are not an artifact of disengaged participants, are provided in the revised manuscript. In addition, please improve the clarity of the methodological details and the results.

I am attaching an Editorial Requests Table that details critical reporting requirements for the revised manuscript. Please attend to each item and ensure your manuscript is fully compliant. If your revised manuscript is not aligned with these requests on major issues, such as those concerning statistics, it may be returned to you for further revisions without re-review.

Please submit the following items:

- Revised manuscript
- Point-by-point response to the referees' comments
- Cover letter (as a separate document)
- <https://www.nature.com/documents/nr-reporting-summary.pdf> Nature Research Reporting Summary
- Completed Editorial Request Table (attached).

via this link: Link Redacted .

Additional guidance is available in our style and formatting guide Communications Psychology formatting guide.

Best regards,

Troby Lui

Troby Lui, PhD
Associate Editor
Communications Psychology

REVIEWER EXPERTISE:

Reviewer #1: deepfake, misinformation

Reviewer #2: deepfake, misinformation

Reviewer #3: deepfake, misinformation, qualitative coding

REVIEWER REPORTS:

Reviewer #1 (Remarks to the Author):

I have enjoyed reading your work and see this as a strong contribution to an ongoing conversation about how we ought prepare for and be thinking about increasingly sophisticated deepfakes.

I have some comments meant to help you as you continue to polish the manuscript.

The intro sounds a little clunk, just in terms of readability. Remove the words 'in fact' in the first sentence, exchange the word 'could' for 'can' in the second sentence. See how that reads...

Do you want to add the words "In spite of Transparent Warnings" to the title? .. it's a lot of words, but your work is unique in the focus narrowly on people who were warned.

Innocuous is likely the wrong word choice in paragraph two - maybe another way to say less nefarious or something...

Can you check for small things? There's a para. without a period at the end, and that sort of thing.

You've been incredibly thorough, study design is very strong and impressive.

Instead of 'as per' how about 'as in the first experiment'.

This idea of a generalized cynicism is important to continue to explore, I hope you will continue this line of work. Important and timely!

Reviewer #2 (Remarks to the Author):

This paper examines online participants' judgments about whether a fictional character is guilty of a crime or moral

transgression based on whether a warning appears before the video saying it is a deepfake. The claim is that people who believe a video is a deepfake and therefore false still believe that the video can be incriminating. This is novel and of interest to the community and wider field. This nicely builds on work in the psychology of misinformation and deepfakes, and the work is convincing with a well-powered randomized experiment and diverse (though small) stimuli. The paper will influence thinking in the field and should be accepted after minor revisions, but it suffers from the fictional nature of the entire exercise. Specifically, there is a “real video” showing an actor confessing to a crime and a “deepfake video” showing an actor confessing to a crime. But both videos are of actors who never actually committed a crime. This is the nature of experiments and the nature of addressing theoretical constructs, so it’s a reasonable limitation, but the fully fictional nature of this is a limitation of this research because the truth is the actor didn’t commit the crime.

The paper (and paragraph 3 in the discussion) shows that people (1) are told a video is a deepfake (2) believed the video is a deepfake (3) still rely on the content of the video to make a judgment of the person’s guilt. What I think would be helpful would be to add a fourth step: Do people recognize the contradiction in their thinking in believing a video is a deepfake yet pointing to the video as evidence for their judgment of guilt? If yes, then all the evidence stands. If no, then this seems to be an artifact of MTurk and Prolific workers giving little attention to this experiment... I would encourage authors to add this fourth step before publishing to offer the strongest evidence for their findings.

I suggest the authors engage with Wittenberg et al 2024 in PNAS Nexus “Labeling AI-Generated Media Online” which states in the abstract: “labels we tested significantly decreased participants’ belief in the presented claims. However, in both studies, labels that simply informed participants that content was generated using AI tended to have little impact on respondents’ stated likelihood of engaging with their assigned post.”

Reviewer #3 (Remarks to the Author):

I congratulate the authors on submitting empirical findings from a much needed kind of study to assess the real world impact of deepfakes (and deepfake technology broadly). The findings, especially the inefficacy of generic content warnings in the fight against people’s perusal of video content (and potentially forming/updating their beliefs as a result) is extremely important. In general, the way I understood the main thesis of the paper (taking cues from the discussion as well) that none of generic content warnings, specific warnings/labels identifying certain media as (deep)faked, media literacy efforts alone will not suffice in preventing harm from deepfakes. Most notably, the authors emphasize the point at many places in the manuscript, and with good reason, that generic warnings might prove to be detrimental to video as a medium writ large, which will be somewhat tragic (my words) considering all that photographic technologies have helped us achieve over the last 200 years (almost!). I commend the authors on taking this study very close to the finish line.

However, there are a few shortcomings in the manuscript that, if addressed sufficiently, will improve it.

- First of all, a matter of convenience: there were no page numbers or line numbers in the manuscript, so it is difficult to provide feedback and be 100% sure that my specification of page numbers will be interpreted correctly. I request the authors to add page numbers in the revised manuscript. I’m using the PDF’s page numbers in the remainder of these comments.
- Generally speaking, I can see the link between “transparency” and providing warnings or content labels, I do not think there is sufficient synthesis in the manuscript to claim that “transparency is insufficient” to counteract deepfakes’ negative influence. I’d argue that for transparency to be a virtue, in the sense the authors wish to employ it, it should be an attribute of the content *creators* rather than the platforms. In my reading, the authors are imputing transparency to the platform (in this case the survey platform, but ostensibly YouTube or Instagram or TikTok in real life) by having it label the content or provide warnings to users. I want the authors to engage just a little bit with this tension to clarify their use of transparency.
- on p. 4, the authors say “Normatively, people should simply disregard the content of a video which they know is fake”. While I agree with the authors at the superficial level, but this assumption does not hold true for entire genres of fiction and entertainment videos that can be created using deepfake tech, and yet carry useful knowledge/wisdom about the world. I’d like to see the authors attempt to qualify this statement.
- The authors’ decision to produce their own deepfake videos is a major strength of this paper that sets this work apart from many others. Good work!
- on p. 7, the authors should list the hypotheses with their respective labels. I understand that the hypotheses are (indirectly) stated in the supplementary files and on the OSF page, but it is my strong recommendation that the authors include them in the main manuscript.
- I substantiate my previous comment by pointing out that while the three studies are innovative, the reporting of the findings is *extremely* difficult to follow. Or at the very least, that was my experience of it. Having the findings section for each experiment be accompanied by the hypothesis labels would have made a patterned and orderly reading possible. As it is now, it was very difficult for me to follow the multitude of statements about the proportions of participants that responded this or that way.
- On p. 7 again, do the authors mean they used CloudResearch’s MTurk Toolkit?
- Same page: the authors say that the participants were “predominantly” based in the US. Unfortunately this is insufficient. At a bare minimum I’d like to see this information tabulated or stated in the prose with respective proportions.
- Again, on p. 7, the authors state their final N’s “after exclusions” based on criteria (stated later). I request the authors to first report how many total responses they collected before deploying their exclusion filters. Without that info, it is hard to obtain a frame for the sample reported here. In other words, this information would enable readers to better understand the authors’ data cleaning strategy.
- In the same vein, it would be helpful to see one short sentence about the attention check deployed.
- Relatedly, in the exclusion criteria on p. 7, the authors state that participants who failed to complete *both* survey items were excluded. Does that mean those who answered only one of the items were retained? If so, I don’t see that fact being

addressed at all in the reporting that follows. Please address this point.

- I was a little confused by the somewhat sudden shift to the word “falsity” to refer to the fakeness (for the lack of a better word) of a video. I request the authors to re-evaluate if “falsity” is indeed a good word to use here, and if so, to please foreshadow it earlier in the manuscript.
- On p. 8, the authors say that a brief explanation was provided to participants unfamiliar with the term ‘deepfake’. How was it determined whether a participant was familiar with the term or not? This needs to be clarified in the manuscript. Also, what do the authors think was the impact of this (un)awareness on the findings? Further, how many participants were unfamiliar with the term?
- I’m unsure about the meaning of the final sentence on p. 8 (“Finally,...”); please consider revising.
- On p. 9, the authors state: “Ratings were combined with a simplified version of the... quantitative data...”. What were the two cut-off points on the 7-point scale to determine the 3 categories (maybe, probably, and definitely)?
- This applies to all the reporting done for the 3 experiments: each time a subgroup count and percentage is stated, there is no mention of the basis (denominator) for that subgroup. As such it is both unconventional, and at the same time quite inconvenient to follow the various percentages stated. I would also request the authors to avoid using the italicized capital N to refer to subgroups/subsamples as a matter of convention and ease of interpretation.
- On p. 11, for experiment 2, the authors state that guilt perception for ‘fake none’ > for ‘fake generic’. Doesn’t that mean that the generic warnings work for fake videos?
- I found it challenging to interpret the numeric reporting for the qualitative responses. I think these would be best interpreted when also summarized in a table.
- While figures 4 and 5 summarize the quantitative findings, I would recommend replacing the bar plots with boxplots because the latter would give a distribution of the ratings (especially when using a jitter variable in your program of choice for generating the plots). You could add lines for mean scores separately (the boxplots by themselves would only indicate the medians). This suggestion is only to emphasize the point that the visuals can be much more informative by giving readers a view of the distribution of user ratings for “falsity” and “guilt” under the various conditions.
- Another point, and I’m not sure if this is because of the submission platform’s rendering or an attribute of the original figures: please use vector graphics (SVG) instead of JPEG or PNG for your figures/diagrams. Because, currently, some of the text is very tiny, and appears quite pixelated when zoomed in.

On the whole, I would also request the authors, that they should include the extended results (on p. 4 of the supplementary file) in the main manuscript, i.e., the responses to suitability and authenticity variables. Because I do think that the suitability (inverted to unsuitability in the supplementary file) measure enriches the picture you’ve tried to paint in the main text. Similarly, the insignificant finding for authenticity should be included in the main manuscript, especially because in my opinion, the fact that some participants made reference to the audio component signals media literacy competencies. This can make your main discussion rather rich and generative for future work. I understand that the OSF page and the supplementary file act as additional spaces for the information reported, but I think that a published report should *at least* summarize all the pertinent information in the main manuscript.

Again, this study is a great step forward in illustrating the contested efficacy of content warnings and labels. However, please do address the above comments as I believe that would genuinely improve the reporting of your work. I look forward to reading the revised version.

Communications Psychology is committed to improving transparency in authorship. As part of our efforts in this direction, we are now requesting that all authors identified as ‘corresponding author’ create and link their Open Researcher and Contributor Identifier (ORCID) with their account on the Manuscript Tracking System prior to acceptance. ORCID helps the scientific community achieve unambiguous attribution of all scholarly contributions. You can create and link your ORCID from the home page of the Manuscript Tracking System by clicking on ‘Modify my Springer Nature account’ and following the instructions in the link below. Please also inform all co-authors that they can add their ORCIDs to their accounts and that they must do so prior to acceptance.

If you experience problems in linking your ORCID, please contact the Platform Support Helpdesk.

Version 1:

Decision Letter:

Dear Mr Clark,

Your manuscript titled "Seeing is Believing: The Continued Influence of AI-Generated 'Deepfake' Videos Despite Transparency Warnings" has now been seen by our reviewers, whose comments appear below. In light of their advice I am delighted to say that we are happy, in principle, to publish a suitably revised version in Communications Psychology.

We therefore invite you to revise your paper one last time to address the remaining concerns of our reviewers and a list of editorial requests. At the same time we ask that you edit your manuscript to comply with our format requirements and to maximise the accessibility and therefore the impact of your work.

EDITORIAL REQUESTS:

Please make sure that reviewer #2's remaining concern regarding the awareness of the contradiction is thoroughly addressed in the revised manuscript. Please note that we will send the work back to the referee to vet your responses and revisions unless, after editorial evaluation, we consider these fully addressed.

SUBMISSION INFORMATION:

OPEN ACCESS:

*** TRANSPARENT PEER REVIEW:** Communications Psychology uses a transparent peer review system. On author request, confidential information and data can be removed from the published reviewer reports and rebuttal letters prior to publication. If you are concerned about the release of confidential data, please let us know specifically what information you would like to have removed. Please note that we cannot incorporate redactions for any other reasons.

*** CODE AVAILABILITY:** All Communications Psychology manuscripts must include a section titled "Code Availability" at the end of the methods section. We require that the custom analysis code supporting your conclusions is made available in a publicly accessible repository at this stage; please choose a repository that generates a digital object identifier (DOI) for the code; the link to the repository and the DOI must be included in the Code Availability statement. Publication as Supplementary Information will not suffice.

*** DATA AVAILABILITY:**

Link Redacted

Best regards,

Troy Lui

Troy Lui, PhD
Associate Editor
Communications Psychology

REVIEWERS' COMMENTS:

Reviewer #1 (Remarks to the Author):

The authors have made excellent edits and have taken all comments from three reviewers very seriously. I am very pleased with all edits.

Reviewer #2 (Remarks to the Author):

This is so close, but I still think there's an important stone left unturned on the "4th step" that I mentioned in my first review: "Do people recognize the contradiction in their thinking in believing a video is a deepfake yet pointing to the video as evidence for their judgment of guilt?"

It is unclear whether we're seeing (1) the results of a double "fictive pass" indicating a "seeing is believing" cognitive bias where if people see something even if they know it's fake they will believe it or (2) the results of people being confused given the many layers of fiction and their inattention.

What's going on here is not simply a standard fictive pass because there's a fictional world in a fictional world in this experiment. The first fictional world is what the authors write allows the fictive pass (e.g. "participants to evaluate moral transgressions by fictional characters much as they would in the real world") but the second fictional world is the fact that there's a deepfake in the fictional world. This is a bit confusing.

Let's consider Ocean's Eleven. If I see Ocean and his crew rob the Bellagio, but I know it's just a hollywood movie, do I (or participants in an experiment) think George Clooney stole from the Bellagio. No. But we all agree the character Ocean is guilty. If I see Ocean's 15, an imaginary sequel, and I see what someone says is a deepfake of Ocean robbing the Bellagio, then I don't know for sure if Ocean is guilty. If I think it's 100% a deepfake in the film, then I shouldn't treat it as evidence for Ocean robbing the Bellagio. But, if I think it's anything less than 100% (and there's good reason because perhaps someone said it's a deepfake but it's not or even if there are glitches perhaps that's just a weird glitch that has nothing to do with AI generation), then it's reasonable for me to assign some likelihood of guilt. But, in any case, my concern is less with this logic and more with the simple point that authors claim "Participants in our experiments were likewise asked to assess guilt based on the content of the video they viewed, regardless of its fictional nature" and I suspect that participants were not paying close enough attention to the fact that the deepfake is fiction inside fiction.

Once again, I suggest minor revisions adding this fourth step in to see if people recognize their contradiction or not. Sorry for asking for additional data, but the complexity of the fiction within fiction makes for an easy alternative explanation of the findings as a result of confusion and lack of attention.

Reviewer #3 (Remarks to the Author):

I thank the authors for addressing my comments in the original review. I believe the manuscript stands much more firmly on its own than the previous version, and on the whole I recommend it for acceptance.

On the notion of transparency, though, I broadly accept the authors' arguments as stated in the rebuttal letter, but I believe that their 'control' condition, is the kind of obstruction of transparency that I was originally referring to. i.e., it's already in the experiment design, but that aspect could truly enrich the discussion. I'm not suggesting that the manuscript is any worse off without reflection on that interpretation of transparency (as in what is visible/audible *in* the video), but I'd like to see the authors follow that line of inquiry in subsequent, perhaps, qualitative study they wish to do in the future. It might yield deeper answers to questions about specific aspects of (perceived) fakeness that tilt people's interpretation of videos.

Relatedly, on p. 21 of the revised manuscript, the authors say that people explained fakeness by referring to specific aspects of the video rather than the warning (despite being warned). The authors reason that "[this] suggests a lack of trust in transparency warnings, or at the very least a lack of trust in our specific warning". But, this could also mean that the power of suggestion (from the warning) played a role in making them notice specific aspects of the videos. i.e., the participants may not be acknowledging the warning as a reason for their skepticism, but the warning might have been the motivator behind that skepticism.

Again, the above are just optional things that the authors might consider addressing in the final manuscript as per their best judgment, the current manuscripts looks good enough to me. Thank you again for engaging in the conversation.

Response to Reviews for COMMSPSYCHOL-25-0270

**Seeing is Believing: The Continued Influence of AI-Generated
'Deepfake' Videos Despite Transparency Warnings**

Simon Clark and Stephan Lewandowsky

Reviews are reproduced in full, in black text. Our responses are in blue text.

Editor

Comment: Editorially, we consider it crucial that evidence to support the data quality for online participants, such as attention checks or additional experimental evidence that the findings are not an artifact of disengaged participants, are provided in the revised manuscript. In addition, please improve the clarity of the methodological details and the results.

Response: Thank you for this opportunity to resubmit our manuscript. We have made a number of major revisions to improve the clarity of our methodological details and results, as detailed below in response to reviewers' comments. You will see that we have also provided additional evidence and clarity in relation to the data quality for online participants.

Reviewer #1

Comment: I have enjoyed reading your work and see this as a strong contribution to an ongoing conversation about how we ought prepare for and be thinking about increasingly sophisticated deepfakes. I have some comments meant to help you as you continue to polish the manuscript.

Response: Thank you for such a positive endorsement of our work and its impact.

Comment: The intro sounds a little clunk, just in terms of readability. Remove the words 'in fact' in the first sentence, exchange the word 'could' for 'can' in the second sentence. See how that reads...

Response: Thank you for these suggestions, which we have implemented. The revised manuscript reads:

“Advances in artificial intelligence (AI) have made it easier to create highly realistic deepfake videos, which can appear to show someone doing or saying something they did not do or say. Deepfakes may present a threat to individuals and society: for example, deepfakes can be used to influence elections by discrediting political opponents.”

Comment: Do you want to add the words "Inspite of Transparent Warnings" to the title? .. it's a lot of words, but your work is unique in the focus narrowly on people who were warned.

Response: This is a helpful observation, and we agree that it's a good idea to edit our title, to emphasize the unique focus of this research. The revised title reads:

“Seeing is Believing: The Continued Influence of AI-Generated ‘Deepfake’ Videos Despite Transparency Warnings”

Comment: Innocuous is likely the wrong word choice in paragraph two - maybe another way to say less nefarious or something...

Response: We have edited the revised manuscript, which reads:

“Although it is possible to create deepfakes for less nefarious purposes such as education and wellbeing, much of the scholarly discussion about deepfakes has focused on the more sinister implications of the technology...”

Comment: Can you check for small things? There's a para. without a period at the end, and that sort of thing.

Response: Thank you for pointing out the lack of a full stop at the end of paragraph 3, which we have now corrected. We have also thoroughly proofread the manuscript again to check for similar typographic errors.

Comment: You've been incredibly thorough, study design is very strong and impressive.

Response: Thanks again for your positive endorsement of our study design.

Comment: Instead of 'as per' how about 'as in the first experiment'.

Response: We have made this change each time the revised manuscript refers back to an earlier experiment, for example:

“Experiment 2 employed a 2 (*real, fake* video) × 3 (*none, generic, specific* warning) between-subjects factorial design, plus a *control* condition as in Experiment 1, making seven conditions in total.”

Comment: This idea of a generalized cynicism is important to continue to explore, I hope you will continue this line of work. Important and timely!

Response: We absolutely agree! The threat of generalized cynicism around AI-generated video content has significant societal implications (e.g., lack of trust and engagement in political discourse, legal implications for video evidence). Video has been described in moral philosophy as an essential “epistemic backstop” (Rini, 2020), on which trust in our experience of the world is grounded. Psychological science can provide a greater understanding of how people respond to AI-generated content, and this will continue to be the focus of our work.

Thank you for your positive review and all of the helpful suggestions for improvement.

Reviewer #2

Comment: This paper examines online participants' judgments about whether a fictional character is guilty of a crime or moral transgression based on whether a warning appears before the video saying it is a deepfake. The claim is that people who believe a video is a deepfake and therefore false still believe that the video can be incriminating. This is novel and of interest to the community and wider field. This nicely builds on work in the psychology of misinformation and deepfakes, and the work is convincing with a well-powered randomized experiment and diverse (though small) stimuli. The paper will influence thinking in the field and should be accepted after minor revisions ...

Response: Thank you for such a positive assessment of our work's quality and impact.

Comment: ... but it suffers from the fictional nature of the entire exercise. Specifically, there is a "real video" showing an actor confessing to a crime and a "deepfake video" showing an actor confessing to a crime. But both videos are of actors who never actually committed a crime. This is the nature of experiments and the nature of addressing theoretical constructs, so it's a reasonable limitation, but the fully fictional nature of this is a limitation of this research because the truth is the actor didn't commit the crime.

Response: We agree that all experiments which employ a fictional scenario will suffer from this limitation, and have accordingly added the following text to our discussion, in the revised manuscript:

"A second limitation common to experiments employing a fictional scenario is the possibility that participants may not respond as they would in a real-world situation (Cook & Campbell, 1979). For example, a participant might conclude that no one is guilty of a crime simply because they are aware that the video features an actor and that no actual crime has occurred. However, our qualitative data provided no indication that this limitation affected participants' reasoning in the present study. Rather, the fictional scenario appeared to confer what has been termed a 'fictive pass', allowing participants to evaluate moral transgressions by fictional characters much as they would in the real world (Thompson et al., 2023). We would argue that the task is analogous to evaluating whether a fictional character in a film is guilty of a depicted transgression: few viewers would conclude that the character is innocent solely on the basis that the film is fictional. Participants in our experiments were likewise asked to assess guilt based on the content of the video they viewed, regardless of its fictional nature."

Comment: The paper (and paragraph 3 in the discussion) shows that people (1) are told a video is a deepfake (2) believed the video is a deepfake (3) still rely on the content of the video to make a judgment of the person's guilt. What I think would be helpful would be to add a fourth step: Do people recognize the contradiction in their thinking in believing a video is a deepfake yet pointing to the video as evidence for their judgment of guilt? If yes, then all the evidence stands. If no, then this seems to be an artifact of MTurk and Prolific workers giving little attention to this experiment... I would encourage authors to add this fourth step before publishing to offer the strongest evidence for their findings.

Response: Thank you for highlighting this potential contradiction in participants' reasoning. The 'fourth step' you suggest is already covered by our conditional analyses, which include only those participants who were shown a warning and subsequently stated that they believe the video is (or may be) a deepfake, indicating that they took note of the warning (or found some other reason to question the video, e.g., technical defects). Even amongst this subset of participants, perception of guilt was higher than in the control groups, indicating that knowing a video is fake does not entirely negate its influence. This supports similar findings by Hughes et al. (2022) who stated that *"neither awareness nor detection serves to protect people from its influence"*. We therefore respectfully question the assertion that our findings only stand if participants recognize the contradiction in their thinking: recognition of one's own contradictions is not a necessary precursor to such contradictions existing – just as people do not necessarily recognize their own cognitive biases. Our conditional analyses demonstrate that people are influenced by the content of a deepfake video, even when they have been told and believe that it is fake. This finding stands, regardless of whether participants are aware of their own contradiction.

Comment: I suggest the authors engage with Wittenberg et al 2024 in PNAS Nexus "Labeling AI-Generated Media Online" which states in the abstract: *"labels we tested significantly decreased participants' belief in the presented claims. However, in both studies, labels that simply informed participants that content was generated using AI tended to have little impact on respondents' stated likelihood of engaging with their assigned post."*

Response: Thank you for bringing this article to our attention. It supports our conclusions, and also the work of Carrella et al. (2025) and Lieb et al. (2024), which we have already cited in our discussion. We have therefore added the following text to our revised manuscript:

"Relatedly, a recent study found that labels informing users that online content was AI-generated reduced belief in that content, but had little effect on users' stated likelihood of engaging with it (Wittenberg et al., 2025), suggesting that such labels may not meaningfully alter behavior. This was demonstrated by our conditional analysis..."

Thank you for such a positive review and your helpful recommendations.

Reviewer #3

Comment: I congratulate the authors on submitting empirical findings from a much needed kind of study to assess the real world impact of deepfakes (and deepfake technology broadly). The findings, especially the inefficacy of generic content warnings in the fight against people's perusal of video content (and potentially forming/updating their beliefs as a result) is extremely important. In general, the way I understood the main thesis of the paper (taking cues from the discussion as well) that none of generic content warnings, specific warnings/labels identifying certain media as (deep)faked, media literacy efforts alone will not suffice in preventing harm from deepfakes. Most notably, the authors emphasize the point at many places in the manuscript, and with good reason, that generic warnings might prove to

be detrimental to video as a medium writ large, which will be somewhat tragic (my words) considering all that photographic technologies have helped us achieve over the last 200 years (almost!). I commend the authors on taking this study very close to the finish line.

However, there are a few shortcomings in the manuscript that, if addressed sufficiently, will improve it.

Response: Thank you for your positive and thoughtful assessment of our manuscript, for recognizing the importance of this research, and for taking the time to produce such thorough and detailed feedback, which we have addressed as follows.

Comment: First of all, a matter of convenience: there were no page numbers or line numbers in the manuscript, so it is difficult to provide feedback and be 100% sure that my specification of page numbers will be interpreted correctly. I request the authors to add page numbers in the revised manuscript. I'm using the PDF's page numbers in the remainder of these comments.

Response: Apologies for not including page numbers in our first manuscript, which was an oversight. Page numbers have been added to the revised manuscript.

Comment: Generally speaking, I can see the link between “transparency” and providing warnings or content labels, I do not think there is sufficient synthesis in the manuscript to claim that “transparency is insufficient” to counteract deepfakes’ negative influence. I’d argue that for transparency to be a virtue, in the sense the authors wish to employ it, it should be an attribute of the content creators rather than the platforms. In my reading, the authors are imputing transparency to the platform (in this case the survey platform, but ostensibly YouTube or Instagram or TikTok in real life) by having it label the content or provide warnings to users. I want the authors to engage just a little bit with this tension to clarify their use of transparency.

Response: This is an excellent point, and aligns with comments made by another reviewer. In stating that transparency is insufficient, we are using the word “transparency” in the same way it is employed by legislators and regulators of online content. For example, the EU Artificial Intelligence Act (2024) refers to “transparency obligations” including a requirement to “disclose that the content has been artificially generated or manipulated”.

This is subtly distinct from a creator being transparent about their content (e.g., in relation to sponsorship or product placement). The two concepts could overlap if a creator disclosed their own use of AI to produce content, but this does not apply to the fictional scenario of our experiments, in which a malicious actor has used AI to create content without the subject’s permission.

We agree that transparency imputed to the platform is materially different to transparency from other sources (e.g., creator, fact checkers, government agency, police) and are planning a future study to test how people respond to warnings from different sources. With this in mind, the revised manuscript reads:

“Our qualitative responses show that a majority of participants who were shown a deepfake video, and were warned beforehand that it was fake, nevertheless commented on some aspect of the video itself (e.g., technical defects) to explain why they believed it to be fake, rather than simply stating that they had been told it was fake. This suggests a lack of trust in transparency warnings, or at the very least a lack of trust in our specific warning, which appeared to have been given by the social media platform. A future study might explore how the source of such a warning (e.g., content creator, social media platform, independent fact checker, police, government) might influence trust in its accuracy.”

Comment: on p. 4, the authors say “Normatively, people should simply disregard the content of a video which they know is fake”. While I agree with the authors at the superficial level, but this assumption does not hold true for entire genres of fiction and entertainment videos that can be created using deepfake tech, and yet carry useful knowledge/wisdom about the world. I’d like to see the authors attempt to qualify this statement.

Response: Again, this is an excellent point, and one to which we have given a great deal of thought in conceptualizing and designing our experiments. We absolutely agree that computer generated imagery (e.g., the movie ‘Avatar’) can convey useful wisdom about the real world. Similarly, our participants might think it plausible (or even typical) that a politician might take a bribe, or a vegan influencer might eat meat, in the real world, and we are planning a future study to test this ‘gist plausibility’. But our experiments ask a direct question about a specific person’s guilt, and it remains the normative response to answer such a question based on evidence – just as we would expect a jury member to base their verdict purely on the evidence presented (rather than, for example, a stereotypical view of the defendant). A deepfake video which we know is fake does not provide any meaningful evidence. [Please also see our response to Reviewer #2, above, in relation to the limitations of studies which employ a fictional scenario.]

We nevertheless agree that our statement on p. 4 was too categorical, and required qualification. The revised manuscript therefore reads:

“Normatively, people should be expected to discount the content of a video they know to be fake – at least when forming beliefs about factual matters – because it has no evidential connection to reality.”

Comment: The authors’ decision to produce their own deepfake videos is a major strength of this paper that sets this work apart from many others. Good work!

Response: Thank you for this positive endorsement of our work. The decision to produce our own deepfake videos for this project was made because we recognized the significant shortcomings of earlier deepfake research which mostly used satirical deepfake videos already in the public domain. Although video production added significantly to the cost and complexity of our research, we agree that it sets this work apart from many others.

Comment: on p. 7, the authors should list the hypotheses with their respective labels. I understand that the hypotheses are (indirectly) stated in the supplementary files and on the

OSF page, but it is my strong recommendation that the authors include them in the main manuscript.

I substantiate my previous comment by pointing out that while the three studies are innovative, the reporting of the findings is extremely difficult to follow. Or at the very least, that was my experience of it. Having the findings section for each experiment be accompanied by the hypothesis labels would have made a patterned and orderly reading possible. As it is now, it was very difficult for me to follow the multitude of statements about the proportions of participants that responded this or that way.

Response: We have addressed this by adding Table 1, which includes descriptions and analysis of all preregistered hypotheses. In the revised manuscript we refer to Table 1 when introducing our hypotheses in the introduction, and again in the results section for each experiment. We have also added hypothesis numbers in brackets throughout the results section, to annotate analyses. For example:

“Perception of fakeness ratings by condition are shown in Figure 5, and analysis of associated hypotheses is shown in Table 1. Perception of fakeness was greater in the fake specific condition (1.27; *maybe yes*) than in the control condition (−0.07; *impossible to say*), $t(156) = 6.07, p < .001, d = 0.93, 95\% \text{ CI } [0.61, 1.25]$ (H2).”

Comment: On p. 7 again, do the authors mean they used CloudResearch’s MTurk Toolkit? Same page: the authors say that the participants were “predominantly” based in the US. Unfortunately this is insufficient. At a bare minimum I’d like to see this information tabulated or stated in the prose with respective proportions.

Response: Our apologies for not being sufficiently clear in describing recruitment; we have clarified both points in the revised manuscript, which reads:

“Participants were recruited from the United States, via the CloudResearch MTurk Toolkit” for Experiments 1 and 2, and “Participants were recruited from the United Kingdom via Prolific” for Experiment 3.

Comment: Again, on p. 7, the authors state their final N’s “after exclusions” based on criteria (stated later). I request the authors to first report how many total responses they collected before deploying their exclusion filters. Without that info, it is hard to obtain a frame for the sample reported here. In other words, this information would enable readers to better understand the authors’ data cleaning strategy.

Response: We have reported total responses and exclusions for each experiment in the revised manuscript. For example:

“From 338 total responses in Experiment 2, 63 participants were excluded: 14 for failing to complete the survey, and 49 for failing the attention check.”

Comment: In the same vein, it would be helpful to see one short sentence about the attention check deployed.

Response: We have described the attention check for each experiment, in the revised manuscript. For example:

“...the attention check required participants to correctly answer the question: “*Where did he say the money is hidden?*”, which was revealed in the closing seconds of the video.”

Comment: Relatedly, in the exclusion criteria on p. 7, the authors state that participants who failed to complete both survey items were excluded. Does that mean those who answered only one of the items were retained? If so, I don’t see that fact being addressed at all in the reporting that follows. Please address this point.

Response: Thank you for pointing out this ambiguity. To clarify, only participants who completed all survey items for the dependent variables were included in our analysis. We have rephrased this sentence in the revised manuscript, which reads:

“Data were excluded for participants who failed to complete any of the survey items for the dependent variables, or failed the attention check, or completed the survey in less than one minute longer than it took to watch the video.”

Comment: I was a little confused by the somewhat sudden shift to the word “falsity” to refer to the fakeness (for the lack of a better word) of a video. I request the authors to re-evaluate if “falsity” is indeed a good word to use here, and if so, to please foreshadow it earlier in the manuscript.

Response: On reflection, we agree with your helpful suggestion that “fakeness” is a better word to use (having checked that it appears in the OED), so have replaced “falsity” with “fakeness” throughout the revised manuscript.

Comment: On p. 8, the authors say that a brief explanation was provided to participants unfamiliar with the term ‘deepfake’. How was it determined whether a participant was familiar with the term or not? This needs to be clarified in the manuscript. Also, what do the authors think was the impact of this (un)awareness on the findings? Further, how many participants were unfamiliar with the term?

Response: You are quite right that we should have been clearer on this point. The brief explanation was provided to all participants in the warning conditions. We have clarified this point in the revised manuscript by adding the phrase “for the benefit of”, thus:

“Both warnings were followed by a brief explanation for the benefit of participants unfamiliar with the term ‘deepfake’, as follows...”

Comment: I’m unsure about the meaning of the final sentence on p. 8 (“Finally,...”); please consider revising.

Response: We have rewritten this paragraph for clarity in the revised manuscript, which reads:

“Finally, participants were thoroughly debriefed about the purpose of the experiment. Participants in the *real specific* condition, who had been told that a real video was a deepfake, were shown a different debrief which revealed this deception.”

Comment: On p. 9, the authors state: “Ratings were combined with a simplified version of the... quantitative data...”. What were the two cut-off points on the 7-point scale to determine the 3 categories (maybe, probably, and definitely)?

Response: We aggregated *maybe, probably, and definitely* responses, so the three categories were simply: (1) negative scores, (2) zero, (3) positive scores. The revised manuscript reads:

“Ratings were then combined with a simplified version of the corresponding quantitative data (*maybe, probably, definitely* aggregated) to give a single measure describing both a participant’s judgement and their reason for making that judgement. For example, the response *definitely yes* / “*He admitted that he took the money*” was rated *guilty, because of video* (coded G/V) whereas the response *maybe not* / “*I can’t say based on a fake video*” was rated *not guilty, because of warning* (coded N/W).”

Comment: This applies to all the reporting done for the 3 experiments: each time a subgroup count and percentage is stated, there is no mention of the basis (denominator) for that subgroup. As such it is both unconventional, and at the same time quite inconvenient to follow the various percentages stated. I would also request the authors to avoid using the italicized capital N to refer to subgroups/subsamples as a matter of convention and ease of interpretation.

Response: To address this, we have added the denominator of each subgroup, throughout the revised manuscript. For example:

“Qualitative responses for perception of guilt are summarized in Table 2. This shows that 89.7% ($n = 35$) of the 39 participants in the *real none* condition, who were shown a real video of John Carter admitting to committing a crime without any prior warning, believed him to be guilty based on the video...”

Thank you for also pointing out our failure to observe the convention of using a lower case italicized n to refer to subgroups/subsamples, which we have corrected throughout the revised manuscript, as in the above example.

Comment: On p. 11, for experiment 2, the authors state that guilt perception for ‘fake none’ > for ‘fake generic’. Doesn’t that mean that the generic warnings work for fake videos?

Response: You are quite right to observe that the generic warning used in Experiment 2 was effective at reducing perception of guilt, as described in our discussion:

“Despite having no influence on their belief that the video was a deepfake, the generic warning nevertheless reduced participants’ perception of guilt.”

This tension between the two dependent variables is further explored in our discussion as we introduce the concept of generalized uncertainty, thus:

“Further, our findings suggest that a generic warning may alter people’s interpretation of the content of a video, even when they remain unconvinced that the video is a deepfake. This is perhaps explained by what the authors of an earlier deepfake study termed “generalized uncertainty”, finding that an educational warning about deepfake videos contributed to cynicism and distrust of social media news more generally (Vaccari & Chadwick, 2020).”

So to answer the question directly, generic warnings work in the sense that they reduce the influence of a fake video, but they may also have unintended consequences by reducing the influence of real videos (e.g., video evidence in court). This is why we conclude:

“To avoid contributing to a generalized cynicism about the authenticity of videos, we therefore suggest avoiding altogether the use of generic warnings about deepfake videos.”

Comment: I found it challenging to interpret the numeric reporting for the qualitative responses. I think these would be best interpreted when also summarized in a table.

Response: As you suggest, we have addressed this issue by adding Table 2, which summarizes qualitative results for all three experiments, showing the percentage of participants in each condition who believed the person was guilty, based on the video.

We have also simplified reporting of qualitative findings in our revised manuscript, with reference to the new table. For example:

“Qualitative responses for perception of guilt are summarized in Table 2. This shows that 53.3% ($n = 40$, coded G/V or G/WV) of the 75 participants in the *fake specific* condition believed John Carter was guilty based (entirely or partially) on the content of the video, compared with 12.0% ($n = 12$) of the 100 participants in the *control* condition.”

Comment: While figures 4 and 5 summarize the quantitative findings, I would recommend replacing the bar plots with boxplots because the latter would give a distribution of the ratings (especially when using a jitter variable in your program of choice for generating the plots). You could add lines for mean scores separately (the boxplots by themselves would only indicate the medians). This suggestion is only to emphasize the point that the visuals can be much more informative by giving readers a view of the distribution of user ratings for “falsity” and “guilt” under the various conditions.

Response: Thank you for this helpful suggestion, which aligns with the publication’s submission guidelines. In the revised manuscript, we have replaced the bar charts with violin plots and individual data points to show distribution, superimposed with mean values and confidence intervals. We agree that this improves reporting by enabling readers to see the distribution of participants’ responses.

Comment: Another point, and I’m not sure if this is because of the submission platform’s rendering or an attribute of the original figures: please use vector graphics (SVG) instead of JPEG or PNG for your figures/diagrams. Because, currently, some of the text is very tiny, and appears quite pixelated when zoomed in.

Response: We have embedded figures and tables into the revised manuscript as vector graphics (SVG). These can also be provided as individual graphics for final publication, to ensure quality.

Comment: On the whole, I would also request the authors, that they should include the extended results (on p. 4 of the supplementary file) in the main manuscript, i.e., the responses to suitability and authenticity variables. Because I do think that the suitability (inverted to unsuitability in the supplementary file) measure enriches the picture you've tried to paint in the main text. Similarly, the insignificant finding for authenticity should be included in the main manuscript, especially because in my opinion, the fact that some participants made reference to the audio component signals media literacy competencies. This can make your main discussion rather rich and generative for future work. I understand that the OSF page and the supplementary file act as additional spaces for the information reported, but I think that a published report should at least summarize all the pertinent information in the main manuscript.

Response: As you suggest, we have moved the extended results for perception of suitability and perception of authenticity into the main manuscript. We have also added the following to the discussion section of the revised manuscript:

“Our perception of suitability measure illustrates the potential for real world consequences: participants’ beliefs about John Carter’s suitability for public office were influenced by the deepfake video, despite the warning.”

“The perception of authenticity measure used in Experiment 1 was intended to test whether participants believed the video had been manipulated, but qualitative responses indicated that the question was widely misunderstood. Many participants referred to the sound effects which had been added to obscure incriminating content in the control condition (e.g., definitely yes / “Most of it was bleeped out”). Whilst such responses perhaps signaled media literacy, this was impossible to isolate using this measure. ...”

Thank you for your positive review and detailed feedback, which has helped us to greatly improve the manuscript for publication.

References

- Carrella, F., Simchon, A., Edwards, M., & Lewandowsky, S. (2025). Warning people that they are being microtargeted fails to eliminate persuasive advantage. *Communications Psychology*, 3(1), 15. <https://doi.org/10.1038/s44271-025-00188-8>
- Cook, T. D., & Campbell, D. T. (1979). *Quasi-experimentation: Design and analysis issues for field settings*. Houghton Mifflin.

- Delacre, M., Lakens, D., & Leys, C. (2017). Why psychologists should by default use Welch's t-test instead of Student's t-test. *International Review of Social Psychology*, 30(1), 92–101. <https://doi.org/10.5334/irsp.82>
- Hughes, S., O. Fried, M. Ferguson, C. Hughes, R. Hughes, X. Yao, and I. Hussey. 2021. Deepfaked online content is highly effective in manipulating people's attitudes and intentions. *PsyArXiv Preprints*. <https://doi.org/10.31234/osf.io/4ms5a>
- Leib, M., Köbis, N., Rilke, R. M., Hagens, M., & Irlenbusch, B. (2024). Corrupted by algorithms? how AI-generated and human-written advice shape (dis)honesty. *The Economic Journal*, 134(658), 766–784. <https://doi.org/10.1093/ej/uead056>
- Rini, R. (2020). Deepfakes and the Epistemic Backstop. *Philosophers' Imprint*, 20(24), 1–16. <https://philpapers.org/rec/RINDAT>
- Thompson, J., Teasdale, B., van Emde Boas, E., Budelmann, F., Duncan, S., Maguire, L., & Dunbar, R. (2023). Does believing something to be fiction allow a form of moral licencing or a 'fictive pass' in understanding others' actions? *Frontiers in Psychology*, 14, 1159866. <https://doi.org/10.3389/fpsyg.2023.1159866>
- Vaccari, C., & Chadwick, A. (2020). Deepfakes and disinformation: Exploring the impact of synthetic political video on deception, uncertainty, and trust in news. *Social Media + Society*, 6(1). <https://doi.org/10.1177/2056305120903408>
- Wittenberg, C., Epstein, Z., Péloquin-Skulski, G., Berinsky, A. J., & Rand, D. G. (2025). Labeling AI-generated media online. *PNAS Nexus*, 4(6), pgaf170. <https://doi.org/10.1093/pnasnexus/pgaf170>

**The Continued Influence of AI-Generated Deepfake
Videos Despite Transparency Warnings**

Simon Clark and Stephan Lewandowsky

Reviews are reproduced in full, in black text. Our responses are in blue text.

Reviewer #1

Comment: The authors have made excellent edits and have taken all comments from three reviewers very seriously. I am very pleased with all edits.

Response: Thank you for such a positive endorsement of our work.

Reviewer #2

Comment: This is so close, but I still think there's an important stone left unturned on the "4th step" that I mentioned in my first review: "Do people recognize the contradiction in their thinking in believing a video is a deepfake yet pointing to the video as evidence for their judgment of guilt?"

It is unclear whether we're seeing (1) the results of a double "fictive pass" indicating a "seeing is believing" cognitive bias where if people see something even if they know it's fake they will believe it or (2) the results of people being confused given the many layers of fiction and their inattention.

What's going on here is not simply a standard fictive pass because there's a fictional world in a fictional world in this experiment. The first fictional world is what the authors write allows the fictive pass (e.g. "participants to evaluate moral transgressions by fictional characters much as they would in the real world") but the second fictional world is the fact that there's a deepfake in the fictional world. This is a bit confusing.

Let's consider Ocean's Eleven. If I see Ocean and his crew rob the Bellagio, but I know it's just a hollywood movie, do I (or participants in an experiment) think George Clooney stole from the Bellagio. No. But we all agree the character Ocean is guilty. If I see Ocean's 15, an imaginary sequel, and I see what someone says is a deepfake of Ocean robbing the Bellagio, then I don't know for sure if Ocean is guilty. If I think it's 100% a deepfake in the film, then I shouldn't treat it as evidence for Ocean robbing the Bellagio. But, if I think it's anything less than 100% (and there's good reason because perhaps someone said it's a deepfake but it's not or even if there are glitches perhaps that's just a weird glitch that has nothing to do with AI generation), then it's reasonable for me to assign some likelihood of guilt. But, in any case,

my concern is less with this logic and more with the simple point that authors claim "Participants in our experiments were likewise asked to assess guilt based on the content of the video they viewed, regardless of its fictional nature" and I suspect that participants were not paying close enough attention to the fact that the deepfake is fiction inside fiction.

Once again, I suggest minor revisions adding this fourth step in to see if people recognize their contradiction or not. Sorry for asking for additional data, but the complexity of the fiction within fiction makes for an easy alternative explanation of the findings as a result of confusion and lack of attention.

Response: Thank you for clarifying your concerns about participants' understanding of the 'nested' fictional structure used in our experiments. We sought advice from the editor about how this issue should be addressed, and based on the editor's guidance have made the following changes to our manuscript:

The first time the experimental paradigm is introduced, we have added the following:

"Because these experiments involved fictional scenarios in which a fictional character appeared in a fictional deepfake video, we relied on participants' ability to make a judgement within the narrative frame provided. We note that this 'nested fiction' structure may have introduced ambiguity in how participants interpreted the video as evidence – an issue to which we return in the discussion."

The limitations section of our discussion has been reworked and includes the following:

"A second limitation common to experiments employing a fictional scenario is the possibility that participants may not respond as they would in a real-world situation ^[43]. It is widely understood, however, that people can make moral judgements about fictional characters in movies. For example, few viewers would conclude that a character is innocent of his on-screen crimes solely on the basis that the film is fictional. This has been termed a 'fictive pass', allowing participants to evaluate the actions of fictional characters much as they would in the real world ^[44]. Participants in the present study were likewise asked to assess guilt based on the content of the video they viewed, regardless of its fictional nature.

Our experiments involved the added complication of fictional scenarios in which a fictional character appeared in a fictional deepfake video, and we note that this 'nested fiction' structure may have introduced ambiguity in how participants interpreted the video as evidence. We therefore cannot rule out the possibility that our deepfake videos' influence may partly reflect inattention to the epistemic structure of the task, rather than pure susceptibility to non-probative information. Distinguishing between these two alternative interpretations is an important direction for future research, which should assess participants' meta-awareness of the fictional scenario, in addition to standard attention checks.

However, our qualitative data provided no indication that this limitation affected participants' reasoning in the present study. Participants' free-text explanations engaged directly with the in-scenario evidence and its credibility, and none suggested uncertainty about the nested fictional structure."

This limitation has also been acknowledged in our abstract, which now reads:

“We found that most participants relied on the content of a deepfake video, even when they had been explicitly warned beforehand that it was fake, although alternative explanations for the video’s influence, related to task framing, cannot be ruled out.”

Reviewer #3

Comment: I thank the authors for addressing my comments in the original review. I believe the manuscript stands much more firmly on its own than the previous version, and on the whole I recommend it for acceptance.

Response: Thank you for your positive evaluation of our revised manuscript.

On the notion of transparency, though, I broadly accept the authors' arguments as stated in the rebuttal letter, but I believe that their 'control' condition, is the kind of obstruction of transparency that I was originally referring to. i.e., it's already in the experiment design, but that aspect could truly enrich the discussion. I'm not suggesting that the manuscript is any worse off without reflection on that interpretation of transparency (as in what is visible/audible *in* the video), but I'd like to see the authors follow that line of inquiry in subsequent, perhaps, qualitative study they wish to do in the future. It might yield deeper answers to questions about specific aspects of (perceived) fakeness that tilt people's interpretation of videos.

Relatedly, on p. 21 of the revised manuscript, the authors say that people explained fakeness by referring to specific aspects of the video rather than the warning (despite being warned). The authors reason that "[this] suggests a lack of trust in transparency warnings, or at the very least a lack of trust in our specific warning". But, this could also mean that the power of suggestion (from the warning) played a role in making them notice specific aspects of the videos. i.e., the participants may not be acknowledging the warning as a reason for their skepticism, but the warning might have been the motivator behind that skepticism.

Again, the above are just optional things that the authors might consider addressing in the final manuscript as per their best judgment, the current manuscript looks good enough to me. Thank you again for engaging in the conversation.

Response: Thank you for generously suggesting areas of future research. These are beyond the scope of the current manuscript but will be valuable in the conceptualisation and design of subsequent studies.